# Dynamically tuning friction at the graphene interface using the field effect

Gus Greenwood[1,5], Jin Myung Kim[2,3,5], Shahriar Muhammad Nahid[4], Yeageun Lee[4], Amin Hajarian[3], SungWoo Nam [3] & Rosa M. Espinosa-Marzal [1,2] ✉

Dynamically controlling friction in micro- and nanoscale devices is possible using applied electrical bias between contacting surfaces, but this can also induce unwanted reactions which can affect device performance. External electric fields provide a way around this limitation by removing the need to apply bias directly between the contacting surfaces. 2D materials are promising candidates for this approach as their properties can be easily tuned by electric fields and they can be straightforwardly used as surface coatings. This work investigates the friction between single layer graphene and an atomic force microscope tip under the influence of external electric fields. While the primary effect in most systems is electrostatically controllable adhesion, graphene in contact with semiconducting tips exhibits a regime of unexpectedly enhanced and highly tunable friction. The origins of this phenomenon are discussed in the context of fundamental frictional dissipation mechanisms considering stick slip behavior, electron-phonon coupling and viscous electronic flow.

Friction plays a key role in both natural and engineered systems, dictating the behavior of sliding contacts, impacting the wear of materials, and influencing the flow of fluids across surfaces, among other effects. The ability to control the friction force at the sliding contact relies on understanding the underlying mechanisms. Beyond the passive control of friction through the selection of the design components (e.g. material and roughness), a more recent trend has been to investigate systems whose frictional response can be dynamically tuned in situ, especially as micro- and nanoscale devices become more common. One of the more promising avenues to achieve friction control is through the use of external electric fields that can modulate the properties of lubricants and material surfaces as well as the interactions between them[1,2].

Novel approaches to the design of interacting surfaces are necessary to move past the current state of the art, and 2D materials are a new and excellent material choice based on their high mechanical strength and chemical and thermal stability. Many 2D materials exhibit intrinsically low friction and are additionally interesting as they can be tuned or doped. This doping effect can be achieved in the traditional sense of atomic substitution, through interactions with the material's environment (e.g., adsorbates or substrates), or through the application of external fields. Graphene, the 2D allotrope of carbon, has been widely studied in this context. Specifically, the graphene field effect transistor (FET) is a common device used to dynamically alter the sheet's properties in situ. FET devices are constructed such that a voltage is applied to a gate that is electrically insulated from the material of interest. This leads to an indirect effect controlled by the resulting electric field, instead of the direct application of bias to the surface. This has the additional advantage that short circuits can be avoided, which protects machine components and lubricants. It is important to point out that while traditional FETs are designed to be turned on and off,

[1]Department of Civil and Environmental Engineering, University of Illinois at Urbana-Champaign, Urbana, IL 61801, USA. [2]Department of Materials Science and Engineering, University of Illinois at Urbana-Champaign, Urbana, IL 61801, USA. [3]Department of Mechanical and Aerospace Engineering, University of California, Irvine, Irvine, CA 92697, USA. [4]Department of Mechanical Science and Engineering, University of Illinois at Urbana-Champaign, Urbana, IL 61801, USA. [5]These authors contributed equally: Gus Greenwood, Jin Myung Kim. ✉e-mail: rosae@illinois.edu

graphene FETs exhibit a gradual response due to graphene's unique band structure[3,4].

Previous works have reported the effects of an electric field between an atomic force microscopy (AFM) tip and semiconducting or graphene surfaces−between which a varying bias was directly applied −on adhesion and friction but have focused on revealing the role of water in humid environments[5-7]. Prior research has also investigated the electronic contribution to friction for semi- and superconductors as their carrier density was modified[2,8-11]. Friction between a tip and doped semiconductor substrates was shown to vary with the local carrier concentration; here, a charge depletion (accumulation) resulted in a substantial decrease (increase) in friction[10,11]. While the contribution of electron wind, charge carrier dragging, and fluctuating electric fields was estimated to be negligible for gallium arsenide covered with an oxide layer[11], the charges trapped in the near-surface layers were responsible for the electrostatically-induced adhesion that justified the variation of the friction force[8,11]. These studies demonstrated the relative importance of factors such as charge density and carrier concentration as well as the electrostatic interaction between the two surfaces. Measurements of nanoscale friction on 2D materials with varying in-plane bias have also shown that applying an increasing bias can decrease the measured friction. This has been primarily attributed to a change in the atomic stick-slip behavior and therefore to the frictional dissipation process[12]. A recent investigation of indirect out-of-plane bias to control carrier concentration via an electric field in a semiconducting $MoS_2$ system associated changes in friction to effects on the electron-phonon coupling[13]. These previous studies were performed under ambient conditions, and hence, the effects of water traces on the results cannot be excluded. Similar effects have not been investigated for graphene either under out-of-plane or indirect bias yet, particularly through the lens of controlling graphene's charge density.

In this work, we study the friction at a single asperity nanoscale contact between the graphene surface of graphene FETs and an AFM tip in a dry nitrogen atmosphere, while the doping level of graphene is modulated in situ by changing the potential applied to the device's back gate. In contrast to conducting or insulating contacts, graphene in contact with semiconducting tips exhibits an enhanced and tunable friction sensitive to the charge density in graphene.

## Results

### Fabrication and characterization of graphene FET device

Figure 1a shows the schematic illustrations of a graphene FET consisting of graphene channel, 300 nm-thick $SiO_2$ dielectric layer, and degenerate silicon backgate. Details of sample preparation are described in the Methods section. Raman spectroscopy on the annealed graphene FETs revealed monolayer characteristic peaks ($I_G/I_{2D} < 0.5$) and marginal $D$ peak intensity (Fig. 1b). Transfer characteristics of the fabricated graphene FET devices (Fig. 1c) showed the charge neutrality point (i.e., Dirac point) at ~25 V for this device, with maximum field-effect mobility estimated ~5000 $cm^2$/Vs. Multiple devices with various Dirac points were used and the device's specific Dirac point is noted in text and Figures. A significant shift in Dirac point was not observed in forward and reverse gate sweeping in dry nitrogen. Repeated sweeps in ambient conditions resulted in large hysteresis and irreversible n-doping of graphene FET devices (Supplementary Fig. 1). Hence, doping from oxygen or water molecules in the atmosphere was prevented by maintaining nitrogen

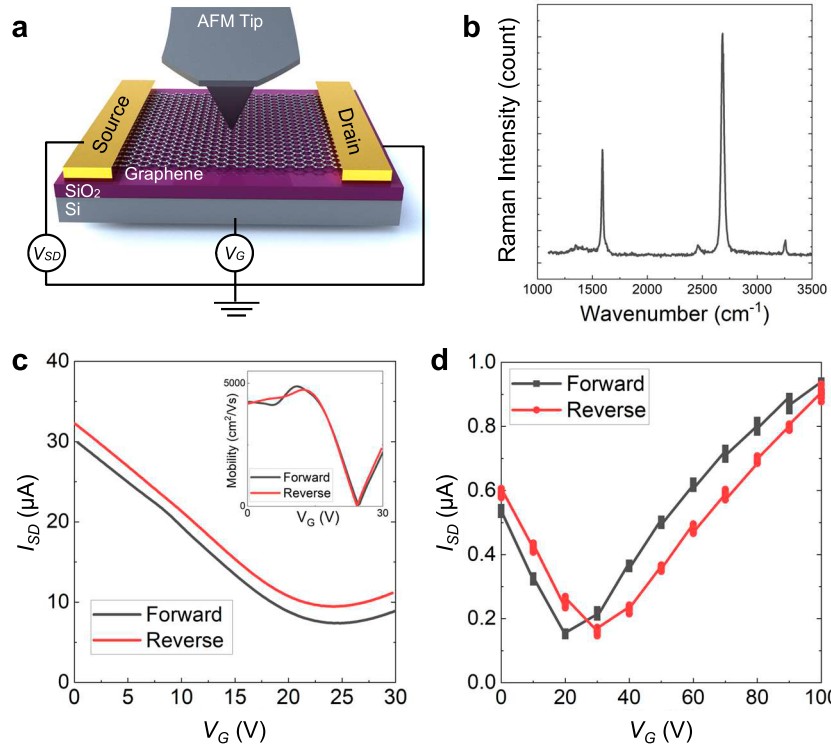

**Fig. 1 | Fabrication and characterization of graphene FET devices used for tunable friction measurements. a** Schematic illustration of in situ electrical characterization of graphene FETs during friction measurement. Note that the tip is not part of the electric circuit, and there is no bias potential applied directly to the tip. The source-drain voltage $V_{SD}$ is maintained constant, while the gate potential $V_G$ is varied. **b** Raman spectrum of a graphene FET channel. **c** Transfer characteristics of graphene FET in dry nitrogen with continuous forward/reverse sweeping of gate bias (with $V_{SD}$= 50 mV). Inset figure shows field-effect mobility of graphene

FETs. **d** Discrete sweeping of gate bias during friction measurements (with $V_{SD}$ = 1 mV). $V_G$ increased from 0 up to 100 V in steps of 10 V and was kept constant for 1 minute at each value for these measurements. The longer duration of these measurements compared to **c** and the higher backgate potential $V_G$ (100 V) increased the probability of interfacial charge transfer between the gate and dielectric (i.e., induced trap charges in the dielectric) during AFM measurements, which caused the small shift of the Dirac point shown in **d**. Source data are provided as a Source Data file.

purging in the measurement cell, which can reduce atmosphere-induced doping as in vacuum. Supplementary Fig. 2 shows the estimated charge carrier density ($n$) as a function of the gate voltage for the example shown in Fig. 1d; here, the electron density at 100 V is $n = 5.69E12$ cm$^{-2}$[14].

AFM was used to measure the adhesion and friction between graphene and AFM tip as well as the non-contact interaction before and after adhesion and friction measurements, both as a function of backgate potential ($V_G$). The backgate potential was increased in steps of 10 V from 0 up to 100 V, kept constant for the same length of time at each value, and then decreased back to zero. During all AFM measurements, the in-plane (i.e., source-drain) potential was held fixed at 1 mV. The AFM tip is not a component of the electrical circuit. That is, there is no potential applied directly to the tip, and hence, the potential between tip and surface is indirectly influenced. In contrast, the charge carrier density (doping) of the graphene sheet is directly controlled in this set up (see Supplementary Fig. 2). Despite purging with dry nitrogen the AFM cell during all measurements, a shift of the Dirac point of -10 V was observed at the end of the experiments, which is consistent with previous reports[15-17], and it implies spontaneous doping due to interfacial charge transfer between the gate and the SiO$_2$.

Three different tip types were used in AFM measurements on the graphene FET devices. Sharp Si tips used directly as supplied by the vendor are referred to as either unmodified or Si tips. Si tips are covered by a thin oxide layer upon exposure to ambient air[18], which is electrically insulating, varies from tip to tip, and can be modified/worn out during contact mode measurements. As described later, these tips behave like n-type semiconductors in terms of charge transfer but can act like an insulating tip before the oxide layer is worn out. The second tip category is gold-coated tips, which are referred to as conducting tips. The gold is electrically conductive, allowing for consistent and repeatable charge transfer to/from the tip when contact is made with the conductive graphene surface. The third type is referenced as insulating tips and consists of either SiO$_2$ (silica) microspheres or thermally oxidized Si tips with a thick oxide layer. Insulating tips do not transfer any charge upon contact with graphene. Reference AFM measurements were carried out on 300 nm SiO$_2$ on Si wafer and 50 nm Au-coated Si wafers, which are labeled as insulating and conductive surfaces, respectively. More information about the experiments is found in the Methods section.

## Charge transfer influences electrostatic interaction

Representative measurements of non-contact forces (before and after friction measurements) are shown in Fig. 2a–c (left column) for the three types of tips. The measured I-V curves indicate that the $V_D$ changes slightly over the course of the experiment, and the width of the yellow bar in each plot reflects this small variation. The non-contact interaction measured with insulating tips (Fig. 2a) increases quasi parabolically with $V_G$ before contacting the graphene surface and is strongly attractive (shown as positive). Such attraction results from the generated electric field, which leads to the polarization of the insulating tip in its proximity; hence, we label this force as electrostatic force, $F_{el}$. The influence of both the insulating tip type (colloid vs. thermally annealed sharp tips) and its size on $F_{el}$ is negligible (Supplementary Fig. 3). The smaller attractive force between tip and graphene compared to the reference silica (SiO$_2$) surface (Supplementary Fig. 4a) indicates that graphene partially screens the electric field. Importantly, the minimum of the parabola is not exactly at $V_G = 0$ V and it shifts during forward and reverse potential leading to a clockwise hysteresis, i.e., $F_{el}$ is smaller during reverse potential sweeps. The hysteresis in $F_{el}$ also occurs on the reference silica surface, and it can be justified by trapped charges forced into the tip and/or silica substrate (underneath graphene in the FET and reference silica) arising from impurities and exposure to the electric field[16,17,19]. Trapped

charges reduce the attractive force and shift the location of the zero tip-sample bias. Figure 2a also shows a small change of $F_{el}$ at each $V_G$ after contacting graphene compared to before contact (labeled as before/after friction). This small change reflects that charge trapping happens upon contact of the insulating tip and graphene.

Figure 2b shows that the use of conductive tips on graphene not only eliminates the parabolic dependence of the electrostatic interaction on $V_G$ but also reduces the attraction significantly, i.e., $F_{el}$ is smaller than 3 nN compared to -100 nN with insulating tips. Note that for the first few gate voltage steps both the insulating and conducting tips experience similar magnitude of $F_{el}$ due to the polarization, but this force becomes approximately constant for the conducting tip once the threshold bias is reached (see inset of Fig. 2b). In contrast, control measurements on the reference silica surface with the same conducting tips showed the unmodified parabolic dependence of $F_{el}$ on $V_G$ and a much higher attraction (Supplementary Fig. 4b). It is thus reasonable that the electric field between conducting tip and graphene is greatly reduced upon contact of the gold-coated tips with graphene due to uninhibited charge transfer between the conductive tip and the current carrying channel. Indeed, there is a smaller difference between $F_{el}$ before and after contact at the beginning of each sweep direction, which suggests that there is a small tip-sample threshold bias necessary to promote charge transfer and reduce the effective electric field even in conducting tips, potentially related to the contact resistance.

The electrostatic attraction between graphene and Si tips (Fig. 2c) exhibits an intermediate response, both in terms of hysteresis between sweep directions and change in behavior after contact with graphene. That is, upon a forward potential sweep, the electrostatic attraction increases initially with $V_G^2$, which reflects the polarizability of the tip, with very small change in the attraction before and after contact with graphene, like the insulating tips. Above a certain $V_G$ value, the difference between the electrostatic attraction before and after contact becomes progressively more pronounced and the electrostatic interaction tends to plateau with a further increase in $V_G$. Both features at high potential are characteristic of semiconducting tips. This transitional behavior (from insulating to conductive) happens at varying potentials depending on the tip and sample (e.g., $V_G$-80 V), but reliably begins at or above the Dirac point $V_D$. The parabolic dependence of $F_{el}$ before and after contact is greatly recovered upon decreasing bias. This indicates that the transition of behavior is not due to wear of the tip's oxide layer, but rather requires crossing some threshold tip-sample bias potentially influenced by the doping of the graphene. Indeed, the difference between the electrostatic attraction before and after contact in the reverse potential sweeps is recovered at very low $V_G$, suggesting a threshold related to the Dirac point, $\Delta V^* = V_G - V_D$. Notably, the minimum of $F_{el}$ in the decreasing sweep is close to $V_D$, as shown for representative measurements. It is, however, noteworthy that the local Dirac point could slightly differ from the macroscopic value obtained from the I-V curve due to subsurface charge-donating impurities[20], potentially justifying the difference between the local minimum of $F_{el}$ and the macroscopic Dirac point (yellow bar). The SI describes the different behavior of the reference samples (Supplementary Fig. 4).

## Friction between graphene and an insulating tip reflects the electric field induced change of the polarization of the tip

Figure 2d displays the frictional characteristics of graphene measured with insulating tips; *cf.* to reference silica and gold surfaces in Supplementary Fig. 5. The $X$ axis shows the potential difference $\Delta V = V_G - V_D$; $V_D$ is the macroscopic Dirac point of graphene, determined via the I–V curve of each graphene sample. The relation between friction ($F$) and $\Delta V$ is parabolic, but it varies across graphene sample/tip pairs. The contact radius, the origin of the silicon oxide (either the surface oxide layer of thermally oxidized tips or the entirely silica microsphere), and the graphene roughness influence the

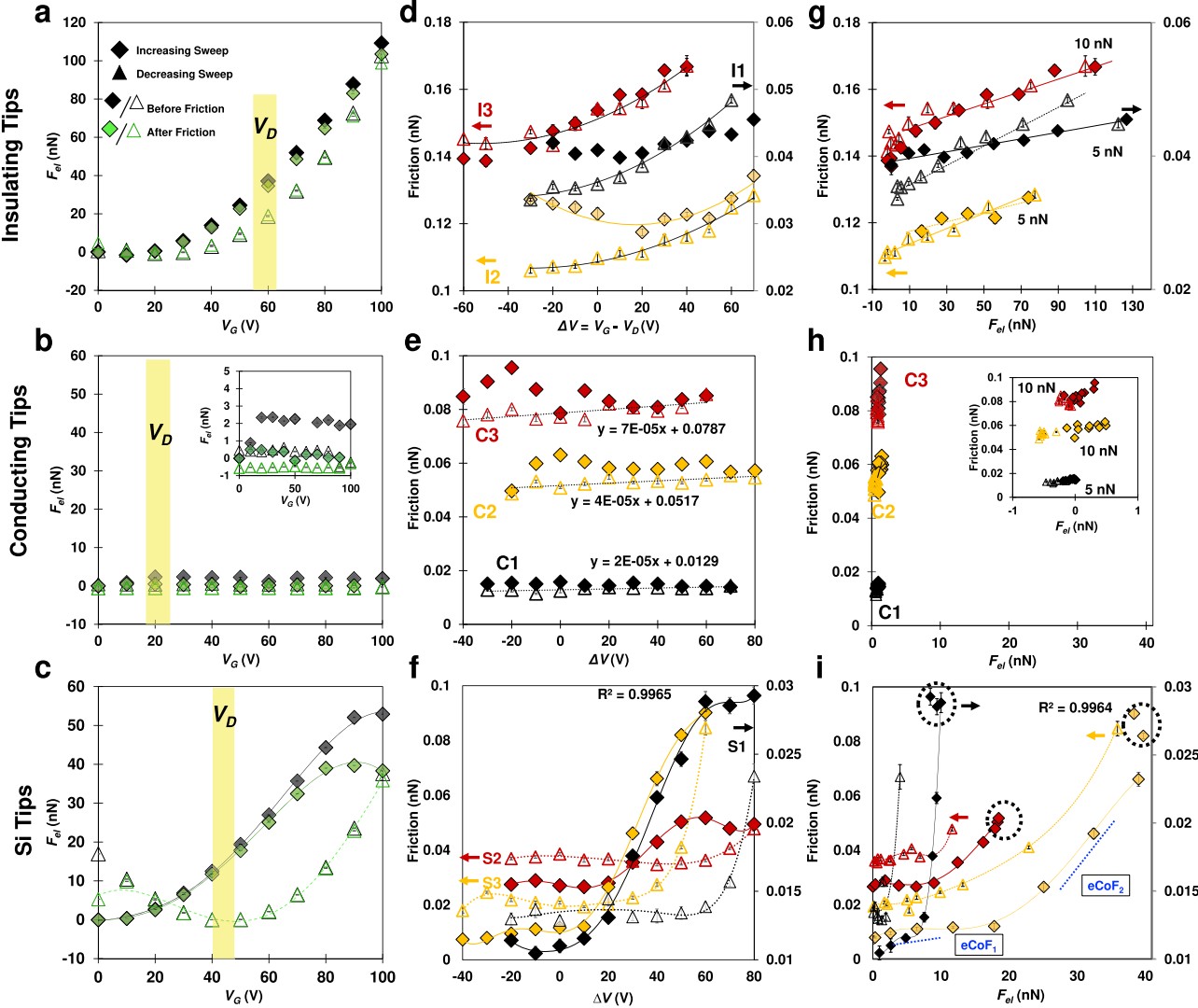

**Fig. 2 | Electrostatic interaction ($F_{el}$) between graphene and insulating, conducting, and Si tip, as well as friction force vs. $\Delta V$ and friction force vs. $F_{el}$.** Each row represents a different tip type. **a**–**c** Representative electrostatic attraction before and after friction measurements as a function of the backgate potential. The legend in **a** applies to the other panels. The yellow bar shows the range of that sample's Dirac point. The width of the bar is meant to indicate local differences and the slight sweep direction hysteresis. The inset of **b** shows a magnified view of the same data; note that $F_{el}$ increases before contact (black diamonds) and remains constant once charge transfer happens. **d**–**f** Friction as a function of the difference between backgate potential and sample's Dirac point, i.e., $\Delta V = V_G - V_D$. **g**–**i** Friction as a function of the non-contact electrostatic attraction measured after friction measurement. Friction plots show results for three graphene/tip pairs labeled as I1-I3, C1-C3, and S1-S3 for insulating, conducting, and Si tips, respectively. **a**, **d**, **g** Results with insulating tips; measurements with thermally annealed tips (I3), and colloidal spheres (I1 and I2) are included. Lines in **d** and **g** are examples of parabolic and linear fits, respectively. **b**, **e**, **h** Results with conducting tips. **c**, **f**, **i** Results with Si tips. Filled and open symbols represent forward and reverse potentials sweeps, respectively, and each set of data points for a forward and reverse sweep represents a different tip/sample pair to demonstrate reproducibility. Error bars, too small to see for many conditions, represent the standard deviation of 5–10 seconds of steady deflection for each condition for electrostatic attraction and the standard deviation of six repeated friction loops for each condition in friction measurements. Source data are provided as a Source Data file.

magnitude of the friction force. Like $F_{el}$, the minimum of the parabola can slightly shift on forward and reverse sweeps yielding a hysteresis. The minima do not collapse on the Dirac point ($\Delta V$=0), indicating that friction is not sensitive to graphene's doping state when using insulating tips, as also observed for the adhesion and $F_{el}$. Hence, this frictional response to $V_G$ is associated with both the tip polarization and the trapped charges, both originating from the electric field. The friction force at graphene/insulating tip contacts scales linearly with the electrostatic attraction, i.e., $F \propto F_{el}$, which is consistent with the relations observed for the electrostatic and friction force: $F_{el} \propto V_G^2$ and that $F \propto V_G^2$. This is also found for the reference SiO$_2$ surfaces (Supplementary Fig. 5a, b). Thus, friction increases while the external load ($L$) is maintained constant, and hence, it can be controlled via an

electric field. For the single asperity contact of our experiments, this can be described as $F = S_{c,V_G = 0} \cdot A + \text{CoF}_{V_G = 0} \cdot L + \text{eCoF} \cdot F_{el}$, where $S_{c,V_G = 0}$ is the critical shear stress, $A$ is the contact area, and CoF is the traditional friction coefficient, which describes the change of friction with load at $V_G = 0$. The third term accounts for the electric field effect via an electronic friction coefficient (eCoF).

## Electrically-induced doping of graphene leads to high frictional dissipation

In the case that the tips are conductive (gold-coated), the variation of friction with $V_G$ is much smaller than with insulating tips, as is also the electrostatic attraction; *cf.* $F$ vs. $V_G$ in Fig. 2e, d. A small anti-clockwise hysteresis was reproducibly observed, with friction upon forward

sweeps being higher than in reverse potential sweeps. A more stable decrease of friction with decreasing $V_G$ is obtained in reverse potential sweeps (-0.00004 ± 0.00002 nN/V) compared to forward sweeps, which may be related to the threshold for charge transfer described earlier. Friction appears to increase with the electrostatic attraction (after contacting graphene) in a very narrow range (Fig. 2h, inset). There is no evidence that friction is sensitive to graphene's doping state. In brief, if charge transfer happens at the graphene/tip contact, the friction control via the electric field is poor.

Friction measurements on graphene with Si tips are summarized in Fig. 2f, i. Friction varies significantly as a function of $V_G$ and $F_{el}$. The differences across pairs of graphene and Si tips (S1-S3) are likely associated with the differences in tip size (R ~ 20 nm for S1 and ~50 nm for S2 and S3, with higher friction values for the latter ones) and graphene roughness. Despite these quantitative differences, the behavior is qualitatively reproducible. First, the hysteresis transitioned from anti-clockwise to clockwise close to the Dirac point of graphene (10-20 V from the macroscopic Dirac point), pointing toward a change of the dissipation at $\sim V_D$. Polynomial functions with orders of 4 describe very well the relation between the friction force and $\Delta V$ (the lines show the fits to the experimental data). Note that the relation between $F_{el}$ and $V_G$ can be also described by a polynomial function of order ~4. However, the relation between friction and $F_{el}$ is not lineal, which implies that friction does not solely originate from the increased electrostatic attraction between a Si tip and graphene; i.e., there must be an additional mechanism underlying the energy dissipation.

When friction is plotted against $F_{el}$, an inflection point is observed in the forward potential sweeps, which invites us to define two different electronic coefficients of friction: eCoF$_1$ and eCoF$_2$ (see blue dashed lines in Fig. 2i to guide the eye). This is a simplification of the reproducible but complex relationship between $F$ and $F_{el}$, but it is useful to compare results for graphene and reference surfaces with different tips. The small friction coefficient eCoF$_1$ characterizes the relation between $F$ and $F_{el}$ below the Dirac point. Above $V_D$, the friction coefficient eCoF$_2$ is much larger. A third regime is also consistently observed for graphene and Si tips, where the friction force either steeply increases or clusters (black dashed circles), coinciding with the plateau of the electrostatic force as a function of potential (Fig. 2i). If the electrostatic force before contact is plot instead of after contact, the clustering vanishes (Supplementary Fig. 6), which indicates that is directly related to the charge transfer in contact.

Reference measurements on silica surfaces with Si and conducting tips show the same trends as with insulating tips (Supplementary Fig. 5a, b): $F_{el} \propto V_G^2$, $F \propto V_G^2$, and $F \propto F_{el}$. This confirms that the polarization of the insulating surface by the electric field dictates the interaction between tip and surface, independently of the conducting characteristics of the tip. When the Si tip slides on gold-coated silicon wafers, the tunability of friction via the electric field is completely lost (Supplementary Fig. 5c, d). The clustering of data points as a function of the electrostatic interaction−as also found on graphene with conductive tips−lets us conclude that the major underlying mechanism, in this case, is the screening of the electric field and reduction of the electrostatic interaction.

**The field-effect induced adhesion between graphene and tip originates from the electrostatic interaction**
The adhesion between graphene and Si/insulating tips qualitatively follows the electrostatic attraction (bottom vs. top plots in Fig. 3),

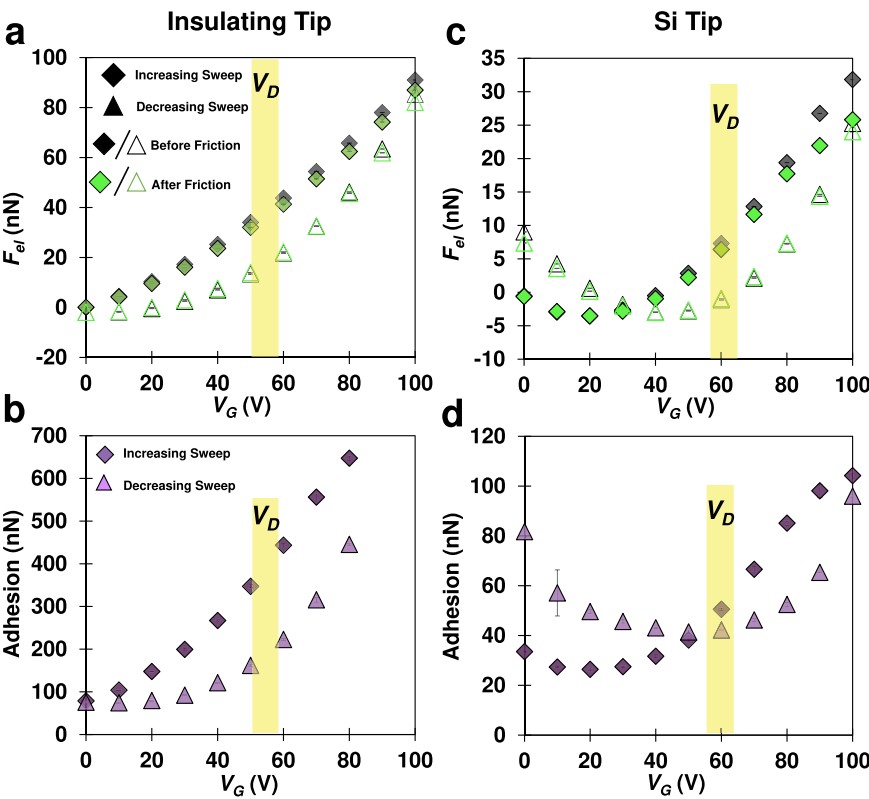

**Fig. 3 | Field-effect induced adhesion between graphene and insulating and Si tips. a, c** Electrostatic interaction and **b, d** adhesion between graphene and **a, b** insulating (thermally annealed with radius R~125 nm) and **c, d** Si tips (R~25 nm). The tip size does not influence $F_{el}$ (Supplementary Fig. 3a). Each point for electrostatic attraction is the average of 5s of steady deflection and for adhesion is the average of five repeated measurements per condition, while error bars give the corresponding standard deviation. The yellow bar in each panel highlights the range of the macroscopic Dirac point of that sample. The electrostatic force increases significantly as the insulating tip becomes closer to the surface (Supplementary Fig. 3b), which justifies the much higher adhesion compared to the Si tip[8]. Source data are provided as a Source Data file.

supporting that the main contribution for the field-effect induced change of adhesion is the electrostatic interaction. Figure 3a, b shows the increase of adhesion with $V_G^2$ measured with insulating tips, with a similar hysteresis between forward and reverse sweeps as the electrostatic interaction. This is consistent with reported adhesion measurements on graphene when the tip-sample bias is directly controlled[5,13]. Figure 3c, d shows the transition of the adhesion between graphene and Si tips from anti-clockwise to clockwise hysteresis, as the electrostatic interaction does, with this transition happening close to the Dirac point.

### Electronic friction coefficient

Figure 4a shows representative measurements of the friction force as a function of load between graphene and three representative tip types at $V_G = 0$ V. The dependence of friction on load is linear in the range of loads investigated for all three tip types, which allows determining a friction coefficient (CoF). The order of magnitude lower friction coefficient with Au-coated tips is attributed to the incommensurability between the lattices of gold and graphene[21]. Figure 4b displays average values for the CoF and the eCoF across multiple experiments for graphene and reference surfaces with each of the three tip types.

When comparing the standard CoF across surfaces, it is clear that graphene has a significantly lower CoF than either of the reference surfaces. The friction coefficient for reference silica and gold surfaces with the three tips is about one order of magnitude larger. The influence of the tip type on the CoF of graphene surfaces is small. In contrast, the measured eCoF significantly depends on the tip behavior. At contacts between graphene and insulating tips, where charge transfer is minimized, friction is primarily dictated by the electric field-induced change of adhesion, and the electronic friction coefficient is obtained from the linear relation $F$ $vs.F_{el}$. The increasing and decreasing sweeps have often similar values, as expected from the minimal hysteresis in these systems. This eCoF (or eCoF$_0$) is roughly half of the CoF, indicating that friction at contacts with minimal charge transfer is more influenced by a varying normal load than a change of electric field. Moreover, this eCoF$_0$ is overestimated based on the magnitude of electrostatic attraction at the distance it is necessarily measured vs. the magnitude in contact (importantly, this is only relevant for the insulating tips and eCof$_0$; see discussion with Supplementary Fig. 3 for details). For friction at graphene/gold-coated tips where charge transfer is maximized there is no eCoF since not only is the change of friction very small but also the adhesion and $F_{el}$ cannot be controlled.

The frictional characteristics of contacts between graphene and Si tips can be modulated by the electric field. For simplification, we define two electronic friction coefficients (eCoF$_1$–labeled low–and eCoF$_2$–labeled high–below and above the Dirac point, respectively). The average value of eCoF$_1$ is comparable to, if not slightly larger than, eCoF$_0$, which indicates that the Si tips act similarly to purely insulating tips in this range. Above the Dirac point, there is a sharp increase in the coefficient of friction (eCoF$_2$ > eCoF$_1$). This value is on average larger than the CoF or eCoF for any other system, suggesting an outsized dependence of friction on the electric field. This frictional response happens when the deviation of the electrostatic attraction before and after contact is observed (Fig. 2c), which suggests a possible influence of charge transfer between surfaces on friction. The largest tunability of the friction coefficient on graphene surfaces via the electric field is thus observed for Si tips.

### Electronic friction excess increases (decreases) with electron (hole) carrier density

Friction measurements were also performed as a function of velocity at multiple backgate potentials. For Si and insulating tips under all conditions, a close to log-linear relationship was obtained between friction and velocity at all backgate potentials; Supplementary Fig. 7 shows representative measurements. To examine the influence of the doping state of graphene, we calculated an excess of friction as $\Delta F = (F - F_D)/F_D$, where $F_D$ is the friction force at the Dirac point of the graphene sample during the forward sweep. Supplementary Fig. 8 shows the calculation at each single velocity for a graphene sample with $V_D$ -50 V. Because $\Delta F$ varies only slightly with velocity, an average value can be determined for each $\Delta V = V_G - V_D$, which is shown in Fig. 5a, with the error bars representing the standard deviation across the range of velocities. Blue colors have been selected to represent the p-doped state $V_G < V_D$, and green colors represent the n-doped state, while black has been chosen for $V_G = V_D$. The change of $\Delta F$ is remarkable. In the p-doped state, friction can decrease up to 60 %, and the minimum value is achieved at $\Delta V = -50$ V. The friction excess raises up to 30% in the n-doped state, pointing toward the influence of both the carrier density and type. By moving further away from the Dirac point, the electron density increases, resulting in higher friction. A decreasing hole carrier density has the same effect as an increasing electron density, it does increase friction. When the potential is reversed, the decrease in the electron carrier density yields a remarkable decrease in friction, but a further reduction of $\Delta F$ in the p-doped state is not seen, suggesting a trapped state has been achieved at the

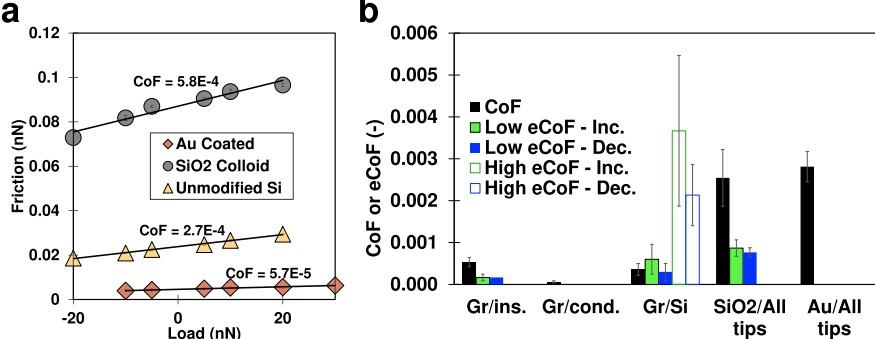

**Fig. 4 | Coefficient of Friction and electronic coefficient of friction.**
**a** Representative load-dependent friction measurement on graphene with a gold-coated tip ($R$-50 nm), a silica colloid ($R$-2.5 μm), and a Si tip ($R$-50 nm), all at $V_G = 0$ V. Solid lines are linear fits with the resulting coefficient of friction reported next to each fit. Error bars are the standard deviation of 8 repeated friction scans in the same location and are smaller than the marker size for these data. **b** Average coefficients of friction (CoF) and electronic coefficients of friction (eCoF) for each

of the three tip types sliding on graphene and on reference silica (SiO$_2$) and gold (Au) surfaces. Error bars are the standard deviation of six or more measurements and include different surfaces and different tips. To determine the eCoF and CoF, at least six experiments were carried out for each combination graphene/tip and at least three experiments for each reference system. Source data are provided as a Source Data file.

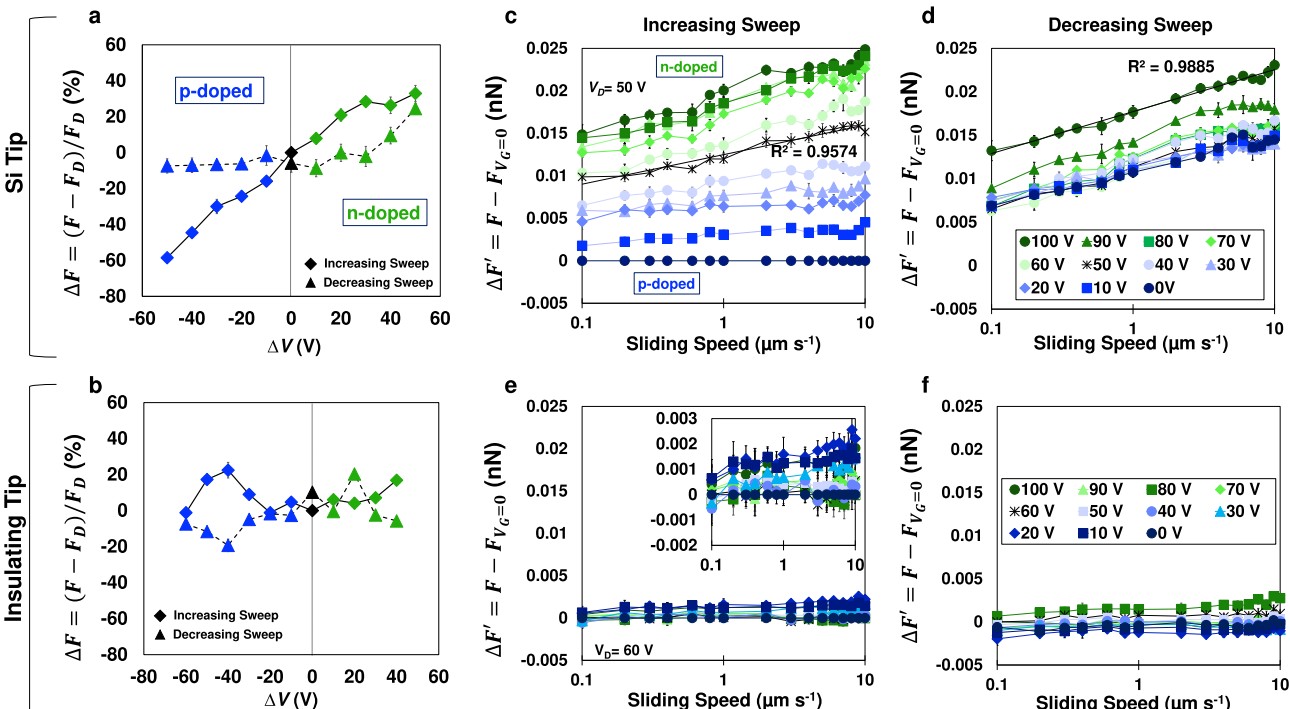

**Fig. 5 | Friction excess for graphene FET devices with Si and insulating tips.**
**a**, **b** Average excess friction $\triangle F$ (%) with respect to the friction force at the Dirac point with Si ($R$ -55 nm) (**a**) and insulating tips (thermally annealed, $R$ -150 nm) (**b**). **c**–**f** $\triangle F'$ vs. sliding velocity measured with Si tip (**c**, **d**) and an insulating tip (**e**, **f**) during a forward potential sweep (**c**, **e**) and a reverse potential sweep (**d**, **f**) on graphene FET devices. Error bars are the standard deviation of six repeated friction sweeps in the same location for all conditions. Source data are provided as a Source Data file.

interface. The release of this trapped state is possible by applying a negative bias[22].

The average $\Delta F$ obtained with an insulating tip is shown in Fig. 5b for comparison, and it is significantly smaller compared to the Si tip. The difference between the p-doped and n-doped regimes is much less significant. This, taken with the maximum friction change of only -20%, reflects that this system is much less sensitive to graphene's doping state. This agrees with the minimal hysteresis, the small eCoF, and overall, the small sensitivity to electric field observed in friction measurements at constant velocity with insulating tips.

To examine the electric field contribution alone, we define a friction excess as $\Delta F' = F - F_{V_G=0}$ at each velocity, where $F_{V_G=0}$ is the friction force at $V_G =$ V. $\Delta F'$ in the decreasing sweep was calculated with respect to the friction force at $V_G = 0$ V before the increasing sweep. Figure 5c, d show $\Delta F'$ for measurements taken with Si tips during forward and reverse sweeps, respectively. During the increasing sweep (Fig. 5c), the friction excess $\Delta F'$ increases with $V_G$. When the potential is reversed, there is a reduction of $\Delta F'$ above the Dirac point (Fig. 5d). However, $\Delta F'$ remains approximately constant when $V_G$ decreases below $V_D$ (see the collapse of the blue curves), consistent with the trapped state described earlier. $\Delta F'$ increases with the logarithm of the sliding velocity up to ~7 µm/s, above which a deviation is observed. Note that Persson[23] proposed that the electronic dissipation in other systems is linearly proportional to the velocity, i.e., $\Delta F' = kV$, where $k$ is a dissipation parameter characteristic of the system. The logarithmic relation between $\Delta F'$ and velocity for graphene clearly deviates from their results. When insulating tips slide on graphene, $\Delta F'$ is very small, and a logarithmic relation between $\Delta F'$ and the velocity is not seen (Fig. 5e, f and inset in 5e).

## Atomic stick-slip reveals modest electric field effects

Figure 6a shows the sawtooth variation of the lateral force at various gate bias, which reveals the stick-slip motion of the Si tip while sliding on graphene. The stick-slip is irregular, displaying a minimum slip length of 2.1 Å, which is close to the lattice spacing of graphene. For these measurements, graphene was deposited on hexagonal boron nitride (hBN)/SiO₂/Si, which leads to smoother surfaces than if directly transferred to SiO₂/Si; see details in Methods. However, the lateral force is still influenced by the surface roughness (RMS ~ 40 pm, and peak-to-valley distance of 350 pm, for a 100 nm² area) and might be partially responsible for the irregular slip features.

The Prandtl-Tomlinson (PT) model for atomic-scale friction of crystalline surfaces attributes the microscopic origin of the energy dissipation to the stick-slip motion of the tip, where the slip length depends on the dimensions of the crystal lattice[24,25]. In this context, the AFM tip is connected by a spring to a moving support that drags the tip with constant velocity over a periodically varying potential landscape $U$, with periodicity corresponding to the crystal lattice. The model explains stick slip as the buildup (as the tip sticks) and then unstable release of energy (when the tip slips), while traversing the periodic energy landscape. The tip stops slipping at the next lattice site (single slip) or at multiples of it[26]. Careful analysis of the stick slip reveals a mixture of single and double slips, with more double slips with increase in $V_G$ and above the Dirac point. Single slips accompany a larger energy loss per unit length than that of multiple slips[27], and hence, they should contribute to the increase of the frictional force with $V_G$. At the same time, the average amplitude of the stick slip, analogous to $U$, appears to have a minimum at the Dirac point and slightly increases as the gate voltage moves away from $V_D$ in either direction (see arrows to the right of Fig. 6a). This minor change does not occur if the Dirac point is not crossed (Supplementary Fig. 9), suggesting that the change in difference between sticking and slipping is related to graphene's changing charge density. Based on this, an additional mechanism must be responsible for the significant increase of the electronic friction coefficient above the Dirac point.

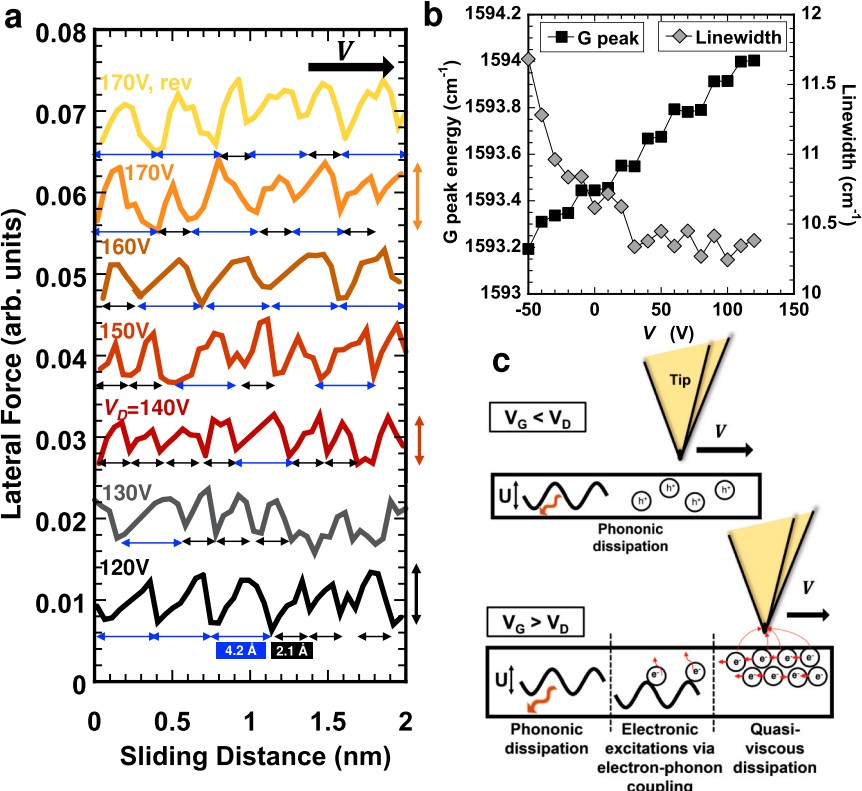

**Fig. 6 | Toward dissipation mechanisms: phononic dissipation, creation of electronic excitation via enhanced electron-phonon coupling, and viscous electronic dissipation. a** Stick-slip behavior of a Si tip sliding on graphene at selected $V_G$ values. A normal load of 10 nN is applied, and the sliding velocity is 20 nm/s. **b** Backgate potential-dependent Raman spectroscopy; the G band energy and linewidth vary with $V_G$, indicating a strengthening of the electron-phonon coupling (see text), and **c** schematics illustrating the proposed dissipation mechanisms. Source data are provided as a Source Data file.

## N-type doping enhances electron-phonon coupling strength

A graphene microFET sample with a Dirac point $V_D \sim 20$ V was used to measure the gate bias-dependent Raman spectroscopy at room temperature; see sample preparation in SI. Figure 6b summarizes the G band position and its linewidth as a function $V_G$; see also Supplementary Fig. 10 for additional information. The G band has been demonstrated to be markedly sensitive to electron-phonon coupling strength[28]. The results show the monotonic increase in G band energy with $V_G$ from −50 V to 125 V ($\sim 0.8\,cm^{-1}$), while the linewidth is reduced ($\Delta\Gamma \sim 1.45\,cm^{-1}$). The smaller linewidth above the Dirac point implies a longer phonon lifetime, which is linked to the increased electron density in n-type doped graphene. While the linewidth quickly plateaus at its reduced value after crossing the Dirac point the monotonic increase of the G band energy continues after crossing the Dirac point; this indicates that a stronger electron-phonon coupling exists in n-doped graphene compared to p-doped graphene and continues to become stronger further past the Dirac point. Note that a previous work showed a symmetric variation of the G band energy and linewidth around the Dirac point, but this was at much lower temperature. At room temperature, asymmetric responses of the G band, similar to our results, have been reported[29].

Both phononic and electronic contributions to friction have been also shown to be temperature dependent[9]. To exclude the influence of thermal effects arising from the applied backgate potential, time-dependent Raman spectroscopy was conducted; experimental details and results are described in the Supplementary Discussion of Supplementary Fig. 11. We conclude that the gate bias does not have a significant effect on sample heating during in situ friction measurements.

## Discussion

In summary, AFM measurements were carried out on graphene FET devices, while the electric field modulated the charge carrier density in graphene. These results demonstrate the prominent influence of the charge transfer across graphene-tip single asperity contacts on the frictional response. When using insulating tips, charge transfer is minimized; the electric field leads to a large electrostatic attraction, and friction scales linearly with this attraction, with a hysteresis due to the effect of trapped charges. Insulating tips thus enable to tune friction via the electric field, in the same way as directly controlling the tip-sample voltage. The electrostatic attraction is eliminated in gold-coated tips due to their conductive nature, allowing charge transfer upon contact, which counteracts the polarization and significantly reduces the dependence of friction on $V_G$. Insulating and conducting tips are not sensitive to graphene's electronic state. In contrast, friction can be dynamically tuned by modulating the doping of graphene in graphene-semiconductor contacts. Here, the electrostatic attraction and adhesion between graphene and Si tips are sensitive to graphene's doping state, as is the frictional response. This has been simplified by introducing a low eCoF and an order of magnitude larger eCoF above the Dirac point. The proposed fundamental mechanisms determining dissipation in this system are illustrated in Fig. 6c and discussed next.

The logarithmic dependence of friction and friction excess on velocity and the features of the stick-slip behavior let us propose an extended PT model, which accounts for the linear superposition of phononic and electronic energy dissipation, and a total potential energy $U_{V_G} = U_{ph} + U_{el,V_G}$, accounting for both contributions, where $U_{ph} = U_{V_G=0}$, and $U_{el,V_G} \neq 0$ for charged graphene, yielding an increase in peak lateral force, amplitude, and periodicity (single to double slip, Fig. 6a). Since only the carrier concentration is changed at constant

load and application of gate voltage does not significantly heat the graphene, $U_{V_G}$ and phononic friction should remain approximately constant under different gate voltages. Note this linear superposition is acceptable because a quasi logarithmic dependence on velocity is also found for the electronic contribution to friction (i.e., friction excess). This extended PT model thus accounts for the damping of lattice vibration (phonons) by creating electronic excitations via electron-phonon coupling, introducing an additional means of dissipating energy[30,31]. Raman spectroscopy (Fig. 6b) supports the enhanced electron-phonon coupling strength with gate bias. Hence, the friction excess could be associated with the stronger electron-phonon coupling. This more efficient dissipation with increasing electron carrier density is likely, at least, a partial contribution to the enhanced friction observed with the Si tips.

An alternative or complementary explanation is based on the description of the charge carrier flow in high-mobility graphene via a hydrodynamic approach[32]. Indeed, electrical transport measurements at finite carrier density reveal a viscous electron flow in graphene that increases with the carrier density around the Dirac point[33,34]. Based on this, we hypothesize that an electronic contribution to friction can originate from the interaction of the AFM tip with the viscous electron (or hole) sea or flow, while sliding. The carrier density within graphene is modulated by $V_G$ (Supplementary Fig. 2); for $V_G > V_D$, it should be an electron sea. The viscosity of the electron liquid, which can achieve very high values up to $0.1\,m^2\,s^{-1}$ at finite carrier density, can be speculated to be analogous to the viscosity of a fluid, hence, to impose a resistance to the motion of the AFM tip, which increases friction, and more so with the increase in sliding velocity. The friction of nanoscale systems in fluids is often interpreted in terms of sliding through a rate-dependent potential energy surface, a model adapted by Bell and Eyring[35–37]. The analysis of $\Delta F'$ using this model is further elaborated in the Supplementary Information (Supplementary Fig. 12). However, we do not have an experimental proof for the correlation between the viscous electron transport in graphene and the frictional dissipation yet. While we suspect this to also be a contributing mechanism to the enhanced friction, future work investigating hydrodynamic descriptions of electron dynamics, particularly in the context of applicability to interactions with external bodies, is necessary for further support.

Looking at electronic effects of a slightly larger scale, recent work also performed using AFM has associated increased friction with large tribocurrents across the sliding interface of chemically modified semiconductors[38]. A study of metal-supported graphene in a conductive AFM demonstrated that current transfer while sliding is associated with higher friction than sliding with no current across the interface, both under an electric field[39]. These authors also observed an enhanced friction not fully explained by changes in the electrostatic attraction. In their fully conductive system, they invoke an electronic property fluctuation model that assumes changes in friction are influenced by both an altered electron density at the interface and the promotion of charge transfer between the sliding surfaces due to the electric field. Similar to the present work, these changes are invoked in the context of the PT model, with both the electron density redistribution and charge transfer leading to an enhanced energy barrier for sliding. Our stick slip measurements do suggest a change of the potential energy landscape when charge transfer occurs, but unfortunately the current across the sliding interface cannot be measured in situ due to the electronically floating semiconducting (Si) tip. So, while we cannot definitively say this for our system, another potential explanation is that the measured excess friction is also influenced by the magnitude and rate of charge transferred between the tip and graphene, which change depending on the tip material, graphene's doping state, and the quantity of induced trapped charges at the interface. The Si tips are more sensitive to these factors than the other two tip types representing the extreme possibilities of charge transfer, and charge transfer can be

encouraged, discouraged, and rate-controlled depending on the charge density in graphene and the material properties of the countersurface, an AFM tip in our case. The interplay between triboelectric currents and the field effect, particularly in semiconducting contacts, thus presents a route for further experimental investigation into electronic contributions to nanoscale friction.

From a device and application perspective, the use of electric field in comparison to electric bias eliminates the need for any direct current flow between the two contacting surfaces and allows energy-efficient operation of friction tuning. Furthermore, our results with graphene also open up opportunities for tunable friction in 2D semiconductor materials-based FET devices where distinct on/off states can be achieved with single polarity charge carriers.

# Methods
## Device fabrication
Graphene FET devices were fabricated by transferring chemical vapor deposition (CVD)-grown graphene to the target substrate. Poly(methyl methacrylate) (abbrev. PMMA, 950A2, Microchem) was spin-coated on graphene (grain size ~5 $\mu m^2$, Sigma Aldrich) on 35 $\mu m$-thick copper foil at 6000 rpm for 30 seconds, followed by baking on a hot plate at 110 °C for 2 minutes. Backside graphene was removed by oxygen plasma treatment (500 W for 10 seconds). Then, PMMA-coated graphene/Cu was floated on a sodium persulfate solution (Reagent grade, >98%, Sigma Aldrich) for 3 hours to completely etch away copper film. The PMMA/graphene was rinsed by immersing into a DI water bath several times and transferred to target substrates, including 300 nm-thick $SiO_2$/Si wafers (UniversityWafer) or metal-coated silicon wafers (e-beam evaporated 50 nm Au/ 3 nm Cr on Si, FC-2000, Temescal). After overnight drying, the PMMA layer was removed by soaking in acetone bath. Graphene was then annealed in a quartz tube furnace at 350 °C for 4 hours under $Ar/H_2$ flowing atmosphere to enhance graphene/substrate adhesion and to eliminate polymer residue on graphene surface. After graphene annealing, source/drain electrodes were prepared using silver paste on both sides of the graphene channel. The fabricated graphene FETs were stored in a vacuum desiccator to prevent degradation and adsorption of water and airborne contamination. The quality of the graphene FET device was confirmed by Raman spectroscopy (Nanophoton Raman 11, Japan) using a 532 nm excitation laser and 2400 l/mm grating. For the electrical characterization, we used a 2-channel sourcemeter (2614B, Keithley). Constant source-drain bias was maintained at 50 mV (for continuous sweeping) or 1 mV (for discrete sweeps), while the gate bias was swept from 0 V to 30 V (in continuous sweeps) or to 100 V (discrete sweeps). The effect of different atmospheres (dry nitrogen and air) on I-V curves of the graphene FET devices was separately examined and compared to vacuum conditions using a vacuum-controlled probe station (FWPX Cryogenic Probe Station, LakeShore).

## Atomic force microscopy (AFM)
AFM measurements were performed with a JPK atomic force microscope (Nanowizard 3, Bruker). All measurements were performed in a constantly refreshed dry nitrogen atmosphere (JPK SmallCell, Bruker) in order to exclude the effects of air humidity and water adsorption. AFM tips had a nominal normal spring constant of 0.3 N/m. Normal spring constants for each cantilever were measured using the Sader's method[40]. Three different tip categories were used in AFM measurements. The first is referred to as Si (unmodified) tips. These are sharp tips as-provided by the manufacturer (HQ:CSC37/NO AL, MikroMasch). The tips are made of n-doped silicon with a typical normal spring constant of ~0.3 N/m and nominal tip radius of 8 nm. The second category of tips is referred to as conducting tips. These are Si tips coated with 5 nm Cr followed by 50 nm Au using an e-beam evaporator (Temescal FC-2000, Ferrotec Corporation). The third tip type is referenced as insulating tip. These are either colloidal silica

microspheres (SS06N, Bangs Laboratories, Inc.) with nominal radius of 5 μm that have been manually glued to tipless cantilevers (HQ:CSC37/TIPLESS/NO AL, Mikromasch) or Si tips that were thermally annealed in ambient atmosphere to induce the growth of a thick oxide layer. There was no significant difference between the AFM results conducted with these two types of insulating tips (see Supplementary Fig. 3 and Fig. 2d, g). The different tip radius of thermally annealed tips (insulating), Si tips, and Au-coated tips (140 nm vs. 20–50 nm vs. 20–70 nm, respectively) influence the magnitude of friction and adhesion, but the results are qualitatively similar. The glue used is a steel-reinforced epoxy (Original Cold-Weld Formula, JB Weld). Images of the investigated regions were collected using both quantitative imaging (QI)/fast force mapping mode and traditional contact mode at low loads ($\leq 5$ nN).

**Friction force measurements.** Friction measurements were performed as a function of the applied load at constant sliding velocity of 1 μm/s and as a function of sliding velocity at constant load (5 or 10 nN), while the backgate bias was maintained constant at a selected value. The gate bias was varied in steps of 10 V from 0 to 100 V, first increasing and then decreasing. All friction scans were performed along a 100 nm line on a flat area of the graphene free from visible contamination. The friction force was calculated from the lateral deflection signal during sliding. A single scan line consists of a trace and retrace line, and friction is calculated by subtracting the retrace from the trace lateral deflection and dividing by two[41,42]. Friction is averaged along each scan line. Ten scan lines are taken at the same conditions and each single data point results from the average of the ten scan lines. The lateral spring constant is estimated using the thermal noise method[43].

**Non-contact force and adhesion.** The measured vertical deflection of the cantilever at a separation from the surface of 3.5 μm yields the non-contact interaction between tip and surface, and as described in Results is of electrostatic origin. The vertical deflection is recorded before the power source applying the gate voltage is turned on. The deflection towards the surface from this baseline is considered a positive value for the electrostatic attraction. Each value is the average of approximately five seconds of measurement of deflection, and the small error bars are the standard deviation over these 5 seconds (Supplementary Fig. 13). Note that since the tip is being constantly attracted while the power source is on, there is an associated drift of the deflection. It becomes more pronounced at higher $V_G$ but is always of smaller magnitude than the overall change due to a change of $V_G$. The cantilever's normal spring constant required to determine the force is measured using the Sader's method[40]. The adhesion between tip and graphene is given by the measured pull-off force (the difference between the out-of-contact baseline and the lowest point on the retraction curve), and the values and error shown are the average and standard deviation for five repeated measurements at each held backgate voltage. Adhesion measurements were performed across multiple tips, samples, and locations on samples to ensure qualitative and quantitative reproducibility.

**Excluding wear of surface and tips.** The wear of both the surface and AFM tips needs to be avoided and/or accounted for to ensure reliable analysis of the measured friction. To exclude the influence of surface wear on AFM results, the investigated areas were imaged both before and after sliding friction; any measurements with changes after measurement in topography, adhesion (images collected in QI mode), or lateral deflection (in contact mode) were not considered. Tip wear (and/or breakage) is also of concern, particularly for sharp Si tips, as unexpected changes in tip geometry and contact area can have noticeable effects on tip-sample interactions. An increase in radius has been previously reported for sharp AFM tips[44] and associated with a one-time fracture event quickly occurring after first contact as

opposed to a continuous wear process. Scanning electron microscopy (SEM) images of an unused tips and of the same tip used only for initial calibration procedures (approach to surface in nitrogen atmosphere and ~5 force curves at a 20 nN or less setpoint) demonstrated that this tip damage, indeed, happened during calibration. Supplementary Fig. 14 shows that the tip noticeably breaks from its pristine ~10–20 nm radius to a radius of ~45 nm or more during calibration. Friction and adhesion measurements after breakage could additionally blunt the tips. However, adhesion measurements with the same tip were also quantitatively reproducible, supporting that any change in tip radius or geometry happened during calibration or imaging of the surfaces before data collection, likely because only very small loads were applied. SEM images of tips used in friction and adhesion measurements had radii between 50 and 75 nm, regardless of the number or type of AFM measurements. Hence, we attribute the increase in tip radius primarily to this initial breaking event so that friction and adhesion data is collected with a consistent tip geometry for all $V_G$ values.

**Atomic stick-slip behavior.** Graphene conforms to $SiO_2$ films leading to RMS roughness ~200–300 pm and peak-to-valley roughness of 1–2 nm for most 100 $nm^2$ areas, which may interfere with the measurement of the atomic stick slip. Hence, graphene/hBN/$SiO_2$ FET devices were fabricated to provide smoother surfaces (typical RMS roughness ~40 pm and peak-to-valley roughness of ~350 pm for a 100 $nm^2$ area on ideal hBN flakes) for stick-slip measurements. First exfoliated hexagonal boron nitride (HQ Graphene) flakes were exfoliated on a clean $SiO_2$/Si wafer using scotch tape. Thin flakes of hBN were found in optical microscope based on the color contrast to the substrate. Target thickness of hBN ranged from 10 to 30 nm, as measured by AFM (note that, depending on the thickness of a given sample, the roughness of the wafer was not always completely removed). Pre-annealing of exfoliated hBN/$SiO_2$ samples was carried out at 400 °C for 3 hours in 50 sccm Ar flow in order to remove the tape residue and improve the adhesion between hBN and $SiO_2$. Then, CVD-grown graphene was transferred on hBN/$SiO_2$ substrate similar to the macrochannel graphene FET fabrication process, described earlier. The presence of hBN underneath graphene resulted in increased number of nanobubbles/blisters trapped at the interface of graphene and hBN, which was observed by dark-field optical microscopy. Subsequent annealing at 350 °C for 4 hours in Ar/$H_2$ atmosphere (Ar 25 sccm, $H_2$ 5 sccm) substantially reduced the density of trapped bubbles in graphene/hBN. Because of the limited areal fraction of hBN flakes to $SiO_2$ substrate, no substantial changes in charge transport characteristics were observed compared to graphene/$SiO_2$ FET devices.

Stick-slip data shown in the main text were collected using the same JPK AFM previously described, while the data shown in the SI were collected using an Asylum Cypher system (Asylum Cypher ES, Oxford Instruments) in lateral force mode, both with purged nitrogen atmosphere. All experimental conditions (e.g., power source connections, gate voltage steps, and timing, calibration procedures) were the same as previously described. The same unmodified sharp tips as previously described were also used for all stick-slip measurements. New tips were used for every new measurement attempt to ensure a radius of 10-20 nm. Suitable hBN flakes were found using each system's optical microscope, followed by minimal AFM imaging of the graphene on hBN (Quantitative Imaging mode in JPK, contact mode in Cypher) to find suitably flat and clean areas. For these measurements in both systems, a 10 nN load was applied while the tip was dragged over a 100 nm scan line at 20 nm/s for 3–5 repeated scans. Stick-slip was found to be more reliably measured at this scan speed using relatively low gain values, the software default for both instruments.

## Data availability

The main text figure data generated in this study are provided in the Source Data file. The raw data and supplementary data sets generated and analyzed in this study are available upon request to the corresponding author. Source data are provided with this paper.

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

## Acknowledgements

We thank J. Yus for help in obtaining SEM images of AFM tips after use and B. Fu for assistance with stick-slip measurements in additional AFM systems. This research was carried out in part in the Materials Research Laboratory Central Research Facilities, University of Illinois. This material is based upon work supported by the National Science Foundation under Grant NSF CMMI-1904216 (R.E.M. and S.N.).

## Author contributions

G.G. performed all AFM and SEM measurements and G.G. and R.E.M. carried out the analysis of AFM data. J.M.K., S.M.N., Y.L., and A.H. performed all device fabrication. J.M.K., S.M.N., and Y.L. aided in the operation of the power source during AFM measurements. J.M.K. S.M.N. and Y.L. performed all characterization of samples (Raman spectroscopy, electrical characterization) and J.M.K. performed the time and gate-dependent Raman measurements. G.G., R.E.M. J.M.K., and S.N. prepared the manuscript and all authors revised. R.E.M. supervised the work of G.G. and S.N. oversaw the work of J.M.K., S.M.N., Y.L., and A.H.

## Competing interests

The authors declare no competing interests.
