## [Peer Review File · Nature Communications]

Dynamically Tuning Friction at the Graphene Interface Using the Field EffectREVIEWER COMMENTS

Reviewer #1 (Remarks to the Author):

In this work, an electric field is applied at graphene/tip interface via a gate bias, and the friction tuning effect of electric field is investigated by the author. The experimental setup and findings are undoubtedly novel and interesting. In the previous works, e.g., [<https://pubs.acs.org/doi/10.1021/acsami.0c06868>] or [<https://doi.org/n0.1038/S41699-022-00316-6>], the bias voltage is applied between the AFM tip and graphene surface, in this case, an electrical current flows across the interface and friction force is tuned by electric current. In this work, the electric current between the tip and graphene is avoided, and the friction force is tuned by electric field, which leads to completely different experimental phenomena and mechanisms. For example, the friction force between a conductive tip and graphene is quite sensitive to electric current [<https://pubs.acs.org/doi/10.1021/acsami.0c06868>], but not sensitive to electric field in this work. Despite the novelty in experimental findings, the underlying mechanisms seem not to be clarified. We appreciate that the authors tried to investigate the mechanisms very carefully, but there are still some important concerns that need to be addressed before I can recommend it to be published.

1. One of the most important findings in this research is the two stages of electronic coefficient of friction (eCoF) of the Si/graphene interface. The authors tried to attribute the difference between eCoF2 and eCoF1 to the electronic dissipation, which is related to the doping state or carrier density of graphene, but they did not give a clear physical picture about how the carrier density influences dissipation and eCoF. Therefore, I suggest the authors to provide a schematic illustration of the electronic friction of Si/graphene interface, which could clarify the relationship between eCoF, electronic dissipation, doping state, carrier density, etc. They can refer to the Figure 9 in [<https://doi.org/10.1080/00018732.2012.706401>]

2. The authors seem to attribute the logarithmic dependence of friction on velocity to the electronic dissipation, i.e., the interaction of the AFM tip with an “electron (or hole) sea” while sliding. They use the Bell-Eyring model, a model that describes the friction of nanoscale systems in fluids, to interpret the logarithmic dependence, which likens the “electron sea” to a fluid. I have two questions about this: 1. The logarithmic dependence of friction on velocity is very common in nanotribology, even if there is no electric field, the friction force between an AFM tip and solids also shows a logarithmic relationship with velocity [<https://journals.aps.org/prl/pdf/10.1103/PhysRevLett.84.1172>]. In this case, how can the authors conclude that the logarithmic dependence is attributed to viscous dissipation of carriers? 2. The electron sea and fluid seem to be completely different things. Electrons follow the Schrödinger equation while fluid follows the Navier-Stokes equation, the author should clarify why they are comparable.

Reviewer #2 (Remarks to the Author):

This paper used three different types of AFM tips: gold-coated conductive tips, Si semiconductor tips and SiO₂ spherical insulator tips to investigate the influence of electric force, adhesion force and friction between tip and graphene under different gate voltages.

The sample used in this article is graphene FET, which backgate voltage and source voltage are equivalent to applying a vertical electric field and a horizontal electric field to the AFM sample. There have been many studies on the application of vertical electric fields, such as these two mentioned articles:

[5] Lang, H., Peng, Y., Cao, X. & Zou, K. Atomic-Scale Friction Characteristics of Graphene under Conductive AFM with Applied Voltages. *ACS Appl Mater Interfaces* 12, 25503-25511, doi:10.1021/acsami.0c06868 (2020).

[a] Electronic Friction and Tuning on Atomically Thin MoS₂. *npj 2D Materials and Applications*, doi:10.1038/s41699-022-00316-6. (2022)

Comparing the experiment of this paper with these two papers, it can be found that the way the backgate voltage is applied and some of the conclusions are similar. For example, several conclusions are mentioned in this article: "Friction between graphene and an insulating tip reflects the electric-field induced change of the polarization of the tip", "Electrically-induced doping of graphene leads to." High frictional dissipation", "The field-effect induced adhesion between graphene and tip originates from the electrostatic interaction", have been mentioned in previous field studies. Specifically, the friction force increases parabolically with the increase of the electric field, and the vertical electric field also increases the interfacial adhesion and electric field force these two conclusions are mentioned before. But the difference is that the experimental environment in this paper excludes the influence of oxygen and water molecules.

Secondly, there are also articles on the influence of horizontal electric field on interface friction, such as:

[1] He F, Yang X, Bian Z, et al. In-Plane Potential Gradient Induces Low Frictional Energy Dissipation during the Stick-Slip Sliding on the Surfaces of 2D Materials[J]. *Small*, 2019, 15(49). DOI: 10.1002/sml.201904613

It is mentioned that the in-plane electric field can be tuned to reduce friction by up to 30%.

Furthermore, this paper investigated the velocity dependence of friction under different electric fields, and its law of close-to-log-linearity is consistent with that without an electric field (specific conclusions are mentioned in nanotribology papers and books). The main part of the viscous dissipation formula under the electric field used at the end of this article is not much different from the previous viscous dissipation formula.

Therefore, the electric field application in this paper can essentially be regarded as the study of the joint application of vertical and horizontal electric fields, but some conclusions have been mentioned in previous field studies. The better point is that it pays attention to the control of the ambient atmosphere.

There are also some issues that need to be improved:

- 1) In the title of Figure 1, why "the higher backgate potential (100 V) increased the probability of interfacial charge transfer"?
- 2) What is the difference between the first type tip before worn out and the third type tip?
- 3) The information in Figure 2 is more cumbersome. There are many data intertwined to make people unclear. It is recommended to redraw the picture to make the reader see more intuitively.
- 4) The change in F_{el} before and after contact in Figure 2a doesn't seem to "be more significant at high V_g " as mentioned.
- 5) Why is there almost no electrostatic force of the conductive tip in Fig. 2b? If the insulator tip has

electrostatic force due to polarization, then why does the conductive tip not have an induced charge and generate electrostatic force?

6) Whether the difference in the diameter of the insulated tips and other tips (μm level and nm class) will affect the electrostatic force and other experimental data?

7) The layout of the article and the pictures also need to be polished.

Reviewer #3 (Remarks to the Author):

This paper presents an experimental study of tunable friction of graphene under electrical field. The results are very interesting and conclusions are adequately supported and discussed. I support its publication after addressing the following comments:

- Can you show some friction trace and retrace under different voltage and velocity? Are you able to see stick-slip friction in the experiments? If so, are the stick-slip patterns varies under different conditions?
- The velocity dependence of friction is analyzed using the Bell and Eyring model. Have you considered thermally-activated Prandtl-Tomlinson (PTT) model. The significance of Prandtl's "time effects" in describing the impact of both sliding speed and temperature on dry friction was recognized by many research recently [Physical Review Letters 106, 126101 (2011); Physical Review Letters 114 (14), 146102, (2015); Physical Review Letters 91, 84502 (2003); Physical Review Letters 96 (2006).]. The attempt frequency should also change under different voltage.
- Is there temperature change during experiments especially at high voltage?

REVIEWER COMMENTS

Reviewer #1 (Remarks to the Author):

In this work, an electric field is applied at graphene/tip interface via a gate bias, and the friction tuning effect of electric field is investigated by the author. The experimental setup and findings are undoubtedly novel and interesting. In the previous works, e.g., [<https://pubs.acs.org/doi/10.1021/acsami.0c06868>] or [<https://doi.org/n0.1038/S41699-022-00316-6>], the bias voltage is applied between the AFM tip and graphene surface, in this case, an electrical current flows across the interface and friction force is tuned by electric current. In this work, the electric current between the tip and graphene is avoid, and the friction force is tuned by electric field, which lead to completely different experimental phenomena and mechanisms. For example, the friction force between conductive tip and graphene is quite sensitive to electric current [<https://pubs.acs.org/doi/10.1021/acsami.0c06868>], but not sensitive to electric field in this work. Despite the novelty in experimental findings, the underlying mechanisms seem not to be clarified. We appreciate that the authors tried to investigate the mechanisms very carefully, but there are still some important concerns need to be addressed before I can recommend it to be published.

1. One of the most important findings in this research is the two stages of electronic coefficient of friction (eCoF) of the Si/graphene interface. The authors tried to attribute the difference between eCoF2 and eCoF1 to the electronic dissipation, which is related to the doping state or carrier density of graphene, but they did not give a clear physical picture about how the carrier density influences dissipation and eCoF. Therefore, I suggest the authors to provide a schematic illustration of the electronic friction of Si/graphene interface, which could clarify the relationship between eCoF, electronic dissipation, doping state, carrier density, etc. They can refer to the Figure 9 in [<https://doi.org/10.1080/00018732.2012.706401>]

The reviewer's suggestion is a good one – we have attempted to describe better the influence of the various mechanisms, but a graphical representation is an important aid in understanding that we originally overlooked.

To rationalize the dissipation mechanisms, we have investigated in detail the stick slip behavior of our system as a function of the backgate potential, and we have also performed bias-dependent Raman spectroscopy. Based on these results, we have also extended the discussion.

We have extended the **discussion of the energy dissipation mechanisms**:

“...The proposed fundamental mechanisms determining dissipation in this system are illustrated in **Figure 6c** and discussed next.

The logarithmic dependence of friction and friction excess on velocity and the features of the stick-slip behavior let us propose an extended PT model, which accounts for the linear superposition of phononic and electronic energy dissipation, and a total potential energy $U_{V_G} = U_{ph} + U_{el,V_G}$, accounting for both contributions, where $U_{ph} = U_{V_G=0}$, and $U_{el,V_G} \neq 0$ for charged graphene, yielding an increase in peak lateral force, amplitude, and periodicity (single to

double slip, Figure 6a). Since only the carrier concentration is changed at constant load and application of gate voltage does not significantly heat the graphene, U_{V_G} and phononic friction should remain approximately constant under different gate voltages. Note this linear superposition is acceptable because a *quasi* logarithmic dependence on velocity is also found for the electronic contribution to friction (*i.e.* friction excess). This extended PT model thus accounts for the damping of lattice vibration (phonons) by creating electronic excitations via electron-phonon coupling, introducing an additional means of dissipating energy.^{1,2} Raman spectroscopy (Figure 6b) supports the enhanced electron-phonon coupling strength with gate bias. Hence, the friction excess could be associated with the stronger electron-phonon coupling. This more efficient dissipation with increasing electron carrier density is likely, at least, a partial contribution to the enhanced friction observed with the Si tips.

An alternative or complementary explanation is based on the description of the charge carrier flow in high-mobility graphene *via* a hydrodynamic approach.³ Indeed, electrical transport measurements at finite carrier density reveal a viscous electron flow in graphene that increases with the carrier density around the Dirac point.^{4,5} Based on this, we hypothesize that an electronic contribution to friction can originate from the interaction of the AFM tip with the viscous “electron (or hole) sea” or flow, while sliding. The carrier density within graphene is modulated by V_G (Figure S2); for $V_G > V_D$, it should be an electron sea. The viscosity of the electron liquid, which can achieve very high values up to $0.1 \text{ m}^2 \text{ s}^{-1}$ at finite carrier density, can be speculated to be analogous to the viscosity of a fluid, hence, to impose a resistance to the motion of the AFM tip, which increases friction, and more so with increase in sliding velocity. The friction of nanoscale systems in fluids is often interpreted in terms of sliding through a rate-dependent potential energy surface, a model adapted by Bell and Eyring.⁶⁻⁸ The analysis of $\Delta F'$ using this model is further elaborated in the Supplementary Information (Figure S12). However, we do not have an experimental proof for the correlation between the viscous electron transport in graphene and the frictional dissipation yet. While we suspect this to also be a contributing mechanism to the enhanced friction, future work investigating hydrodynamic descriptions of electron dynamics, particularly in the context of applicability to interactions with external bodies, is necessary for further support.”

Figure 6. Toward dissipation mechanisms: phononic dissipation, creation of electronic excitation via enhanced electron-phonon coupling, and viscous electronic dissipation. **a**, Stick slip behavior of a Si tip sliding on graphene at selected V_G values. A normal load of 10 nN is applied, and the sliding velocity is 20 nm/s. **b**, Backgate potential-dependent Raman spectroscopy; the G band energy and linewidth vary with V_G , indicating a strengthening of the electron-phonon coupling (see text), and **c**, schematics illustrating the proposed dissipation mechanisms.

2. The authors seem to attribute to the logarithmic dependence of friction on velocity to the electronic dissipation, i.e., the interaction of the AFM tip with an “electron (or hole) sea” while sliding. They use Bell-Eyring model, a model describes the friction of nanoscale systems in fluids, to interpret the logarithmic dependence, which liken the “electron sea” to a fluid. I have two questions about this: 1. The logarithmic dependence of friction on velocity is very common in nanotribology, even if there is no electric field, the friction force between an AFM tip and solids also shows logarithmic relationship with velocity

[\[https://journals.aps.org/prl/pdf/10.1103/PhysRevLett.84.1172\]](https://journals.aps.org/prl/pdf/10.1103/PhysRevLett.84.1172). In this case, how can the authors conclude that the logarithmic dependence is attributed to viscous dissipation of carriers?

The reviewer makes a good point. The original version of the manuscript invokes the Bell-Eyring model by assuming the electron sea has a viscous effect on friction, similar to a lubricated contact. This is based on following findings, which have been now added to the manuscript. There is evidence to support that the flow of charge carriers in charge-neutral graphene resembles that of hydrodynamic fluids (see e.g. Ku, Mark JH, Tony X. Zhou, Qing Li, Young J. Shin, Jing K. Shi, Claire Burch, Laurel E. Anderson et al. "Imaging viscous flow of the Dirac fluid in graphene." *Nature* 583, no. 7817 (2020): 537-541). The measurements let us establish a parabolic Poiseuille profile (instead of the conventional ohmic transport of electrons) for electron flow in a high-mobility graphene channel, establishing the viscous transport of the Dirac fluid at room temperature. The viscous electronic transport has been probed via magnetic field imaging. Electrical transport measurements at finite carrier density are consistent with hydrodynamic electron flow in graphene (Bandurin, D. A. et al. Negative local resistance caused by viscous electron backflow in graphene. *Science* 351, 1055–1058 (2016).) The viscosity of graphene's electron liquid at finite charge carrier density is found to be $\approx 0.1 \text{ m}^2 \text{ s}^{-1}$, larger than at charge neutrality, and an order of magnitude larger than that of honey. These references have been added to the manuscript.

While the viscous flow of electrons in high-mobility graphene over a wide range of charge carrier densities is well established, we do not have evidence for the extension of this hydrodynamic electron transport to frictional dissipation due to viscous effects. And hence, we do not have evidence that these results justify that friction increases only above the Dirac point. This has not been demonstrated yet. Hence, while we still mention this comparison in the Discussion section, we focus on alternative friction mechanisms more extensively now. We have extended the results section with stick slip measurements and the discussion with the Prandtl-Tomlinson model. Furthermore, we introduce the electron-phonon coupling as a potential dissipation mechanism above the Dirac point, which is supported by our stick slip measurements.

The analysis using the Bell and Eyring model has been moved to the SI.

3. The electron sea and fluid seem to be completely different things. Electrons follow Schrodinger equation while fluid follows N-S equation, the author should clarify why they are comparable.

There is evidence to support the transport of charge carriers in high-mobility graphene as a viscous flow. As described by Bandurin et al. the collective behavior of many-particle systems that undergo frequent inter-particle collisions is routinely described by the theory of hydrodynamics. The theory relies only on the conservation of mass, momentum and energy and is highly successful in explaining the response of classical gases and liquids to external perturbations varying slowly in space and time. More recently, it has been shown that hydrodynamics can also be applied to strongly interacting quantum systems including ultra-hot nuclear matter and ultra-cold atomic Fermi gases in the unitarity limit. In principle, the hydrodynamic approach can also be employed to describe many-electron phenomena in

condensed matter physics. The theory becomes applicable if electron-electron scattering provides the shortest spatial scale in the problem, so that electron-electron collisions are sufficiently frequent to provide local equilibrium. Under these conditions, electrons can behave as a viscous liquid and exhibit hydrodynamic phenomena similar to classical liquids. Experiments have probed the validity of the hydrodynamic approach in high-mobility single layer and bilayer graphene. We refer to Bandurin et al.⁴ and references herein for details about the experimental methods. This work also describes the terms included in the proposed Navier Stokes equation, and why some terms have been discarded. One of the implications of such viscous electron transport is the formation of submicrometer-size whirlpools in the electron flow in doped graphene. The viscosity of graphene's electron liquid is found to be $\approx 0.1 \text{ m}^2 \text{ s}^{-1}$, an order of magnitude larger than that of honey, in agreement with many-body theory.

As discussed in the previous point, this section of the text has been edited to explain more clearly the analogy by using appropriate literature. But now, we emphasize the relationship between friction and viscous flow of electrons less strongly, since we do not have evidence for this. We provide an alternative mechanism underlying the excess in friction, supported by experimental results and literature.

Reviewer #2 (Remarks to the Author):

This paper used three different types of AFM tips: gold-coated conductive tips, Si semiconductor tips and SiO₂ spherical insulator tips to investigate the influence of electric force, adhesion force and friction between tip and graphene under different gate voltages.

The sample used in this article is graphene FET, which backgate voltage and source voltage are equivalent to applying a vertical electric field and a horizontal electric field to the AFM sample. There have been many studies on the application of vertical electric fields, such as these two mentioned articles:

[5] Lang, H., Peng, Y., Cao, X. & Zou, K. Atomic-Scale Friction Characteristics of Graphene under Conductive AFM with Applied Voltages. *ACS Appl Mater Interfaces* 12, 25503-25511, doi:10.1021/acsami.0c06868 (2020).

[a] Electronic Friction and Tuning on Atomically Thin MoS₂. *npj 2D Materials and Applications*, doi:10.1038/s41699-022-00316-6. (2022)

Comparing the experiment of this paper with these two papers, it can be found that the way the backgate voltage is applied and some of the conclusions are similar. For example, several conclusions are mentioned in this article: "Friction between graphene and an insulating tip reflects the electric-field induced change of the polarization of the tip", "Electrically-induced doping of graphene leads to." High frictional dissipation", "The field-effect induced adhesion between graphene and tip originates from the electrostatic interaction", have been mentioned in previous field studies. Specifically, the friction force increases parabolically with the increase of the electric field, and the vertical electric field also increases the interfacial adhesion and electric field force these two conclusions are mentioned before. But the difference is that the experimental environment in this paper excludes the influence of oxygen and water molecules. Secondly, there are also articles on the influence of horizontal electric field on interface friction, such as:

[1] He F, Yang X, Bian Z, et al. In-Plane Potential Gradient Induces Low Frictional Energy Dissipation during the Stick-Slip Sliding on the Surfaces of 2D Materials[J]. *Small*, 2019, 15(49). DOI: 10.1002/sml.201904613

It is mentioned that the in-plane electric field can be tuned to reduce friction by up to 30%. Furthermore, this paper investigated the velocity dependence of friction under different electric fields, and its law of close-to-log-linearity is consistent with that without an electric field (specific conclusions are mentioned in nanotribology papers and books). The main part of the viscous dissipation formula under the electric field used at the end of this article is not much different from the previous viscous dissipation formula.

Therefore, the electric field application in this paper can essentially be regarded as the study of the joint application of vertical and horizontal electric fields, but some conclusions have been mentioned in previous field studies. The better point is that it pays attention to the control of the ambient atmosphere.

We thank the reviewer for these comparisons and references and have updated the introduction and description of adhesion to include the references not previously discussed. We also appreciate the reviewer's interpretation of our study and comparison to previous literature, although we do disagree about the conclusion about the novelty and realize that the original text might have been misleading.

We would like to emphasize the main differences with respect to these previous works in our opinion (in addition to the control of the environment):

Lang, H., Peng, Y., Cao, X. & Zou, K. Atomic-Scale Friction Characteristics of Graphene under Conductive AFM with Applied Voltages. ACS Appl Mater Interfaces 12, 25503-25511, doi:10.1021/acsami.0c06868 (2020).

In that work, a bias voltage is applied between the AFM tip and graphene surface and an electrical current flows across the interface and friction force is tuned by electric current. In our work, the tip is not part of the electric circuit, and the friction force is tuned by the electric field, instead, which leads to a different experimental setup and results. For example, the friction force between conductive tip and graphene is quite sensitive to electric current (Lang et al.), but not sensitive to electric field in Lang et al.'s work.

Electronic Friction and Tuning on Atomically Thin MoS₂. npj 2D Materials and Applications, doi:10.1038/s41699-022-00316-6. (2022)

One important difference here is that MoS₂ is a semiconductor and graphene is a semimetal. Another key difference is that the setup in the previous work does not have an electrode on MoS₂, which has to connect to ground for applying a gate bias. Because of the floating configuration, electric field applied to MoS₂ is expected to be not well defined, which is quite different from the well-defined electric field effect that we investigate.

He F, Yang X, Bian Z, et al. In-Plane Potential Gradient Induces Low Frictional Energy Dissipation during the Stick-Slip Sliding on the Surfaces of 2D Materials. Small, 2019, 15(49). DOI: 10.1002/sml.201904613.

In all our experiments, the in-plane bias was held at a fixed value (1 mV), and hence, it was not

used to modulate friction. This is in contrast to He et al.'s study, who investigated the modulation of friction via the in-plane potential.

In response to the reviewer's remark, we have updated the **Introduction** to emphasize the differences from previous works.

“Measurements of nanoscale friction on 2D materials with varying in-plane bias have also shown that applying an increasing bias can decrease the measured friction. This has been primarily attributed to a change in the atomic stick-slip behavior and therefore to the frictional dissipation process⁹. A recent investigation of indirect out-of-plane bias to control carrier concentration via an electric field in a semiconducting MoS₂ system associated changes in friction to effects on the electron-phonon coupling¹⁰. These previous studies were performed under ambient conditions, and hence, the effects of water traces on the results cannot be excluded. Similar effects have not been investigated for graphene either under out-of-plane or indirect bias yet, particularly through the lens of controlling graphene's charge density. In this work, we study the friction at a single asperity nanoscale contact between the graphene surface of graphene-FETs and an AFM tip in a dry nitrogen atmosphere, while the doping level of graphene is modulated *in situ* by changing the potential applied to the device's backgate. In contrast to conducting or insulating contacts, graphene in contact with semiconducting tips exhibits an enhanced and tunable friction sensitive to the charge density in graphene.”

There are also some issues that need to be improved:

1) In the title of Figure 1, why “the higher backgate potential (100 V) increased the probability of interfacial charge transfer”?

Thanks to the reviewer for catching this detail. In this figure we describe the interfacial charge transfer between the gate and dielectric, which leads to trap charges that can shift the measured Dirac point, but this was not clearly described. The Figure 1 caption has been updated to make obvious which interface is being described.

“**Figure 1. Fabrication and characterization of graphene FET devices used for tunable friction measurements.** **a**, Schematic illustration of *in-situ* electrical characterization of graphene FETs during friction measurement. Note that the tip is not part of the electric circuit, and there is no bias potential applied directly to the tip. The source-drain voltage V_{SD} is maintained constant, while the gate potential V_G is varied. **b**, Raman spectrum of a graphene FET channel. **c**, Transfer characteristics of graphene FET in dry nitrogen with continuous forward/reverse sweeping of gate bias (with $V_{SD} = 50$ mV). Inset figure shows field-effect mobility of graphene FETs. **d**, Discrete sweeping of gate bias during friction measurements (with $V_{SD} = 1$ mV). V_G increased from 0 up to 100 V in steps of 10 V and was kept constant for 1 minute at each value for these measurements. The longer duration of these measurements compared to **c** and the higher backgate potential V_G (100 V) increased the probability of interfacial charge transfer between the gate and dielectric (*i.e.* induced trap charges in the dielectric) during AFM measurements, which caused the small shift of the Dirac point shown in **d**.”

2) What is the difference between the first type tip before worn out and the third type tip?

The third tip type has a thick silicon oxide layer (after annealing and oxidation) that is not worn out completely, and hence, it behaves like an insulating tip. We identify this simply by measuring the electrostatic force as a function of the back gate potential, which leads to results as in Figure 1a. The first tip type was not annealed, and hence, the natural oxide layer is very thin, if present. Before this thin layer is worn out the electrostatic force is similar to Figure 1a. After wear, the electrostatic force changes to Figure 1c and remains constant.

3) The information in Figure 2 is more cumbersome. There are many data intertwined to make people unclear. It is recommended to redraw the picture to make the reader see more intuitively.

While we agree that Figure 2 is a busy figure, we think that showing it as in the original manuscript is important to illustrate both the variety of results and reproducibility of the measurements across different tip/sample pairs. However, we recognize the problem and to improve clarity we have modified the caption of Figure 2:

“Figure 2. Electrostatic interaction (F_{el}) between graphene and insulating, conducting and Si tip, as well as friction force vs. ΔV and friction force vs. F_{el} . Each row represents a different tip type. a-c, Representative electrostatic attraction before and after friction measurements as a function of the backgate potential. The legend in (a) applies to the other panels. The yellow bar shows the range of that sample’s Dirac point. The width of the bar is meant to indicate local differences and the slight sweep direction hysteresis. The inset of (b) shows a magnified view of the same data; note that F_{el} increases before contact (black diamonds) and it remains constant once charge transfer happens. d-f, Friction as a function of the difference between backgate potential and sample’s Dirac point, i.e. $\Delta V = V_G - V_D$. g-i, Friction as a function of the non-contact electrostatic attraction measured after friction measurement. Friction plots show results for three graphene/tip pairs labelled as I1-I3, C1-C3 and S1-S3 for insulating, conducting and Si tips, respectively. a, d, g, Results with insulating tips; measurements with thermally annealed tips (I3), and colloidal spheres (I1 and I2) are included. Lines in d and g are examples of parabolic and linear fits, respectively. b, e, h, Results with conducting tips. c, f, i, Results with Si tips. Filled and open symbols represent forward and reverse potentials sweeps, respectively, and each set of data points for a forward and reverse sweep represent a different tip/sample pair to demonstrate reproducibility. Error bars, too small to see for many conditions, represent the standard deviation of 5-10 seconds of steady deflection for each condition for electrostatic attraction and the standard deviation of 6 repeated friction loops for each condition in friction measurements.”

We have also added many clarifications in the manuscript text describing this figure.

4) The change in F_{el} before and after contact in Figure 2a doesn’t seem to "be more significant at high V_g " as mentioned.

Thanks to the reviewer for this accurate interpretation. This line was part of a previous interpretation of the results and is no longer relevant. The manuscript has been updated to reflect this:

“This small change reflects that charge trapping happens upon contact of the insulating tip and graphene.”

5) Why is there almost no electrostatic force of the conductive tip in Fig. 2b? If the insulator tip has electrostatic force due to polarization, then why does the conductive tip not have an induced charge and generate electrostatic force?

Before the charge transfer occurs for the conducting tips the magnitude of F_{el} is quite similar compared to that of the insulator tip. However, once the threshold bias for the conducting tip charge transfer is reached, the electrostatic force remains constant. The magnified inset of Figure 2b demonstrates this, and both the Figure 2 caption and text discussion have been updated to draw attention to it. The reason is that the charge transfer screens the electric field.

“Figure 2b shows that the use of conductive tips on graphene not only eliminates the parabolic dependence of the electrostatic interaction on V_G but also reduces the attraction significantly, i.e. F_{el} is smaller than 3 nN compared to ~ 100 nN with insulating tips. Note that for the first few gate voltage steps both the insulating and conducting tips experience similar magnitude of F_{el} due to the polarization, but this force becomes approximately constant for the conducting tip once the threshold bias is reached (see inset of Figure 2b). In contrast, control measurements on the reference silica surface with the same conducting tips showed the unmodified parabolic dependence of F_{el} on V_G and a much higher attraction (Figure S4b). It is thus reasonable that the electric field between conducting tip and graphene is greatly reduced upon contact of the gold-coated tips with graphene due to uninhibited charge transfer between the conductive tip and the current carrying channel.”

In Figure 2:

“The inset of (b) shows a magnified view of the same data; note that F_{el} increases before contact (black diamonds) and it remains constant once charge transfer happens.”

6) Will the difference in the diameter of the insulated tips and other tips (μm level and nm class) will affect the electrostatic force and other experimental data?

Thanks for the question. This is a very important point. The orders of magnitude difference in size happens for the insulating tips. As you can see in the new Figure S3 (added to the Supplementary Information) the electrostatic interaction is negligibly influenced by the size of the insulating tip ($R \sim 2.5 \mu\text{m}$ vs. 150 nm). This does indeed influence the other experimental data measured in contact, primarily seen in the different magnitudes of friction measured for a given tip-surface combination with different tips. This variation does not affect our conclusions but is important to note, so we have also updated the results and methods to include the radii and discuss the effects.

As also shown in Figure 2d;g, the friction force between graphene and the insulating tips is qualitatively similar when measured with a colloid and a thermally annealed tip. The contact radius between graphene and a silica colloid with a radius of $2.5 \mu\text{m}$ at 10 nN applied normal load is 5.4 nm , whereas it is 2.12 nm for a tip with a radius of 150 nm (a factor of 2 smaller), and hence, one would expect higher friction with the colloid (experiments I1 and I2), compared to the sharp tip (I3). However, asperities on the colloid surface are very common and might lead to smaller contact radii that would justify the results; as described in the manuscript text, graphene roughness also influence these results. However, based on these results, we conclude that the

difference in size among the insulating tips is not a major influencing factor on our experimental data.

We have added following explanation to the SI:

Figure S3. Electrostatic interaction between graphene and insulating tips. The magnitude of F_{el} is determined using the difference between the deflection of the tip at 0 V (baseline uninfluenced by electrostatics) and a given gate voltage both at 3.5 μm from the surface (see Methods and Figure S13). The data are taken from the experiments used in Figure 2 for insulating tips, corresponding to experiments I1, I2, and I3. For simplicity of this comparison only the increasing voltage sweep is shown. The plot compares results taken with a thermally annealed sharp tip with a radius of 150 nm (red triangles) and two colloids with radii of 2.5 μm (black and yellow circles). The electrostatic interaction measured at 3.5 μm from the surface is negligibly influenced by the type and size of the insulating tips. As also shown in Figure 2d;g, the friction force between graphene and the insulating tips is qualitatively similar when measured with a colloid and a thermally annealed tip. The contact radius between graphene and a silica colloid with a radius of 2.5 μm at 10 nN applied normal load is 5.4 nm, whereas it is 2.12 nm for a tip with a radius of 150 nm (a factor of 2 smaller), and hence, one would expect higher friction with the colloid (experiments I1 and I2), compared to the sharp tip (I3). However, asperities on the colloid surface are very common and may lead to smaller contact radii that would justify the results. Based on these results, we conclude that the difference in size among the insulating tips is not a major influencing factor on our experimental data. It does influence the magnitude of friction and adhesion, but the results are qualitatively similar.”

When comparing thermally annealed tips (insulating) with Si tips and gold-coated tips, there is also difference in tip radii, as the reviewer recognizes, since friction and adhesion depend on the contact radius. We have added various explanations in Methods and Results section to emphasize this:

We have added to **Methods**:

“There was no significant difference between the AFM results conducted with these two types of insulating tips (see Figure S3 and Figure 2d;g). The different tip radius of thermally annealed tips (insulating), Si tips and Au-coated tips (140 nm vs. 20-50 nm vs. 20-70 nm, respectively)

evidently influence the magnitude of friction and adhesion, but the results are qualitatively similar.”

In Results:

“Friction measurements on graphene with Si tips are summarized in Figures 2f and 2i. Friction varies significantly as a function of V_G and F_{el} . The differences across pairs of graphene and Si tips (S1-S3) are likely associated with the differences in tip size (R~20 nm for S1 and ~50 nm for S2 and S3, with higher friction values for the latter ones) and graphene roughness.”

To make clear these differences, we have also added the size of the tips used to each figure and emphasize these comparisons in the text. In addition to this, the influence of the tip size (and graphene roughness) on friction are reflected in the error bars of the friction coefficients (Figure 4b).

7) The layout of the article and the pictures also need to be polished.

Thank you for the thorough revision. We have improved the quality and line widths of the figures throughout, in addition to significantly updating the text at the end of Results and into Discussion. We hope that these changes have addressed the reviewer’s request to improve the presentation of our work.

Reviewer #3 (Remarks to the Author):

This paper presents an experimental study of tunable friction of graphene under electrical field. The results are very interesting, and conclusions are adequately supported and discussed. I support its publication after addressing the following comments:

- Can you show some friction trace and retrace under different voltage and velocity? Are you able to see stick-slip friction in the experiments? If so, are the stick-slip patterns varies under different conditions?

This is a good suggestion from the reviewer. We have performed extensive experiments to study the stick slip behavior, as we think these results should aid in the overall interpretation of the enhanced friction. Graphene is deposited on a silicon oxide film which is quite rough and conforms to this rough surface. This interfered with our measurements of atomic stick slip. To improve the quality of the measurements, we deposited graphene on hBN, which is crystalline, but also an insulator like SiO₂ and performed the same type of experiments. In this case, we resolved better the stick slip behavior. The principal conclusion is that the variation of the stick slip features with backgate voltage is not as remarkable as the increase of the lateral peak force above the Dirac point. This analysis has enabled us to improve the discussion of the mechanisms of energy dissipation.

In Methods:

“Atomic stick-slip behavior

Graphene conforms to SiO₂ films leading to RMS roughness ~ 200-300 pm and peak-to-valley roughness of 1 – 2 nm for most 100 nm² areas, which may interfere with the measurement of the atomic stick slip. Hence, graphene/hBN/SiO₂ FET devices were fabricated to provide smoother surfaces (typical RMS roughness ~40 pm and peak-to-valley roughness of ~350 pm for a 100 nm² area on ideal hBN flakes) for stick-slip measurements. First exfoliated hexagonal boron nitride (HQ Graphene) flakes were exfoliated on a clean SiO₂/Si wafer using scotch tape. Thin flakes of hBN were found in optical microscope based on the color contrast to the substrate. Target thickness of hBN ranged from 10 to 30 nm, as measured by AFM (note that, depending on the thickness of a given sample, the roughness of the wafer was not always completely removed). Pre-annealing of exfoliated hBN/SiO₂ samples was carried out at 400°C for 3 hours in 50 sccm Ar flow in order to remove the tape residue and improve the adhesion between hBN and SiO₂. Then, CVD-grown graphene was transferred on hBN/SiO₂ substrate similar to the macrochannel graphene FET fabrication process, described earlier. The presence of hBN underneath graphene resulted in increased number of nanobubbles/blisters trapped at the interface of graphene and hBN, which was observed by dark-field optical microscopy. Subsequent annealing at 350°C for 4 hours in Ar/H₂ atmosphere (Ar 25 sccm, H₂ 5 sccm) substantially reduced the density of trapped bubbles in graphene/hBN. Because of the limited areal fraction of hBN flakes to SiO₂ substrate, no substantial changes in charge transport characteristics were observed compared to graphene/SiO₂ FET devices.

Stick slip data shown in the main text were collected using the same JPK AFM previously described, while the data shown in the SI were collected using an Asylum Cypher system (Asylum Cypher ES, Oxford Instruments) in lateral force mode, both with purged nitrogen atmosphere. All experimental conditions (e.g. power source connections, gate voltage steps and

timing, calibration procedures) were the same as previously described. The same unmodified sharp tips as previously described were also used for all stick slip measurements. New tips were used for every new measurement attempt to ensure a radius of 10-20 nm. Suitable hBN flakes were found using each system's optical microscope, followed by minimal AFM imaging of the graphene on hBN (Quantitative Imaging mode in JPK, contact mode in Cypher) to find suitably flat and clean areas. For these measurements in both systems, a 10 nN load was applied while the tip was dragged over a 100 nm scan line at 20 nm/s for 3-5 repeated scans. Stick slip was found to be more reliably measured at this scan speed using relatively low gain values, the software default for both instruments."

In Results

"Atomic stick-slip reveals modest electric field effects

Figure 6a shows the sawtooth variation of the lateral force at various gate bias, which reveals the stick slip motion of the Si tip while sliding on graphene. The stick-slip is irregular, displaying a minimum slip length of 2.1 Å, which is close to the lattice spacing of graphene. For these measurements, graphene was deposited on hBN/SiO₂/Si, which leads to smoother surfaces than if directly transferred to SiO₂/Si; see details in Methods. However, the lateral force is still influenced by the surface roughness (RMS ~40 pm, and peak-to-valley distance of 350 pm, for a 100 nm² area) and might be partially responsible for the irregular stick slip features.

The Prandtl-Tomlinson (PT) model for atomic scale friction of crystalline surfaces attributes the microscopic origin of the energy dissipation to the stick-slip motion of the tip, where the slip length depends on the dimensions of the crystal lattice^{11,12}. In this context, the AFM tip is connected by a spring to a moving support that drags the tip with constant velocity over a periodically varying potential landscape U , with periodicity corresponding to the crystal lattice. The model explains stick slip as the buildup (as the tip sticks) and then unstable release of energy (when the tip slips), while traversing the periodic energy landscape. The tip stops slipping at the next lattice site (single slip) or at multiples of it.¹³ Careful analysis of the stick slip reveals a mixture of single and double slips, with more double slips with increase in V_G and above the Dirac point. Single slips accompany a larger energy loss per unit length than that of multiple slips,¹⁴ and hence, they should contribute to the increase of the frictional force with V_G . At the same time, the average amplitude of the stick slip, analogous to U , appears to have a minimum at the Dirac point and slightly increases as the gate voltage moves away from V_D in either direction (see arrows to the right of Figure 6a). This minor change does not occur if the Dirac point is not crossed (Figure S9), suggesting that the change in difference between sticking and slipping is related to graphene's changing charge density. Based on this, an additional mechanism must be responsible for the significant increase of the electronic friction coefficient above the Dirac point. "

- The velocity dependence of friction is analyzed using the Bell and Eyring model. Have you considered thermally-activated Prandtl-Tomlinson (PTT) model. The significance of Prandtl's "time effects" in describing the impact of both sliding speed and temperature on dry friction was recognized by many research recently [Physical Review Letters 106, 126101 (2011); Physical Review Letters 114 (14), 146102, (2015); Physical Review Letters 91, 84502 (2003);

Physical Review Letters 96 (2006)]. The attempt frequency should also change under different voltage.

Thank you for the suggestion. In our previous version of the manuscript, we had decided to focus the discussion on the viscosity analogy, and hence, we chose Bell and Eyring model. Based on the feedback from you and the other reviewers, as well as more literature survey, we now also consider the PT model in the discussion, which we connect to the damping of the phononic dissipation by creating electronic excitations due to the higher electron-phonon coupling strength above the Dirac point.

“The logarithmic dependence of friction and friction excess on velocity and the features of the stick-slip behavior let us propose an extended PT model, which accounts for the linear superposition of phononic and electronic energy dissipation, and a total potential energy $U_{V_G} = U_{ph} + U_{el,V_G}$, accounting for both contributions, where $U_{ph} = U_{V_G=0}$, and $U_{el,V_G} \neq 0$ for charged graphene, yielding an increase in peak lateral force, amplitude, and periodicity (single to double slip, Figure 6a). Since only the carrier concentration is changed at constant load and application of gate voltage does not significantly heat the graphene, U_{V_G} and phononic friction should remain approximately constant under different gate voltages. Note this linear superposition is acceptable because a *quasi* logarithmic dependence on velocity is also found for the electronic contribution to friction (*i.e.* friction excess). This extended PT model thus accounts for the damping of lattice vibration (phonons) by creating electronic excitations via electron-phonon coupling, introducing an additional means of dissipating energy.^{1,2} Raman spectroscopy (Figure 6b) supports the enhanced electron-phonon coupling strength with gate bias. Hence, the friction excess could be associated with the stronger electron-phonon coupling. This more efficient dissipation with increasing electron carrier density is likely, at least, a partial contribution to the enhanced friction observed with the Si tips.”

We have not found any information in the literature about how the attempt frequency (of the tip) depends on V_G . Note that the potential is not directly applied to the tip in our experimental set up, but the tip is under the effect of the electric field. We did not find any information about the influence of the electric field on the attempt frequency neither. Nevertheless, the fact that the stick slip characteristics do not significantly change, support that this is not a key factor to consider.

• **Is there temperature change during experiments especially at high voltage?**

Literature (Freitag, Marcus, Mathias Steiner, Yves Martin, Vasili Perebeinos, Zhihong Chen, James C. Tsang, and Phaedon Avouris. "Energy dissipation in graphene field-effect transistors." Nano letters 9, no. 5 (2009): 1883-1888.) suggests that the graphene surface shouldn't experience any significant temperature change, but to confirm this in our system we have also performed additional experiments using Raman spectroscopy during the voltage sweeps.

A brief description has been added to the Results and the further details have been added to the SI.

In Results:

” Both phononic and electronic contributions to friction have been also shown to be temperature dependent.¹⁵ To exclude the influence of thermal effects arising from the applied backgate

potential, time-dependent Raman spectroscopy was conducted; experimental details and results are described in the SI Text and Figure S11. We conclude that the gate bias does not have a significant effect on sample heating during *in situ* friction measurements.”

In the SI:

Figure S11. Time-dependent Raman spectroscopy measurements. **a**, Transfer characteristics of graphene FET in ambient environment with continuous forward/reverse sweeping of gate bias (with $V_{SD} = 50$ mV). **b**, Field-effect mobility of graphene FETs. **c**, Time-dependent Raman spectra. The maps show the G peak (position and intensity) as a function of time and back gate potential. **d**, Time-dependent shift of the G peak over time as a function of the back gate potential. At back gate potentials of 50 and 100 V, there is a peak shift after ~ 50 seconds, which is attributed to trapped charges.

Potential-and time dependent Raman spectroscopy

Both phononic and electronic contributions to friction have been also shown to be temperature dependent.¹⁵ To exclude the influence of thermal effects, time-dependent Raman spectroscopy with a 532 nm CW laser (power = 1 mW) and 1800 lpmm (lines per mm) grating spectrometer (XperRAM, Nanobase, South Korea) was performed on graphene field-effect transistors that exhibited a Dirac point at 10 V (Figure S11a,b). The sample was prepared with commercial graphene (Grolltex) that was O_2 plasma treated right before graphene transfer. This may cause p-doping, yielding a low Dirac point. The sample was thermally annealed at 450 °C. The Raman signal was collected at a fixed position every 20 seconds until the total accumulated time reached 5 minutes. The gate bias was varied from -30 V (p-doping) to 100 V (n-doping), and we kept the gate bias constant for each set of measurements. The measurements were carried out in ambient environment.

Because the G peak exhibits a temperature-sensitive shift of $-0.016 \text{ cm}^{-1}/\text{K}$,¹⁶ the heating of the sample surface should be reflected in a gradual negative shift of the G peak. Figure S11c displays the G peak position and its intensity as a function of measurement time, while the G peak shifts are summarized in Figure S11d. The results show no correlation between the G peak position and the time under gate bias. We only observed an abrupt positive shift in the case of n-type doping ($V_G = 50 \text{ V}, 100 \text{ V}$), but the peak became saturated in less than 1 min. This shift is attributed to the increase in surface charge traps under ambient environment.¹⁷ It is possible that the heating leads to a small negative shift ($<-0.3 \text{ cm}^{-1}$) that is not detected with the precision of the spectrometer, implying a temperature increase $<20 \text{ K}$. Because the influence of the temperature on atomic scale friction is sublinear,¹⁸ this could not explain the significant increase in the friction force above the Dirac point. Although hot-carrier induced heating of graphene devices has been reported earlier, this occurred by Joule heating under high source-drain bias and direct current flow.^{19,20} For instance, a source-drain bias of 1 V can increase the temperature of a graphene FET device by 100 K, with maintaining a linear relationship between source-drain bias and temperature increase up to 1000 K.⁸ Our friction measurements were performed under low source-drain bias ($V_{SD}=1 \text{ mV}$). Although we applied high gate bias, the power in the graphene channel is limited to several nW, which can exclude heating. We also note that, if there were less than 100 K temperature increase, it would not affect the doping level or carrier density in graphene FET, since doping-induced carrier greatly exceeds the thermally induced graphene at room temperature. Thus, we conclude that the gate bias does not have a significant effect on sample heating during *in situ* friction measurements. ”

References

- 1 Filleter, T. & Bennewitz, R. Structural and frictional properties of graphene films on SiC(0001) studied by atomic force microscopy. *Physical Review B* **81**, 155412 (2010).
- 2 Lee, H., Ko, J. H., Choi, J. S., Hwang, J. H., Kim, Y. H., Salmeron, M. & Park, J. Y. Enhancement of Friction by Water Intercalated between Graphene and Mica. *J Phys Chem Lett* **8**, 3482 (2017).
- 3 Ku, M. J. H. *et al.* Imaging viscous flow of the Dirac fluid in graphene. *Nature* **583**, 537 (2020).
- 4 Bandurin, D., Torre, I., Kumar, R. K., Ben Shalom, M., Tomadin, A., Principi, A., Auton, G., Khestanova, E., Novoselov, K. & Grigorieva, I. Negative local resistance caused by viscous electron backflow in graphene. *Science* **351**, 1055 (2016).
- 5 Mayzel, J., Steinberg, V. & Varshney, A. Stokes flow analogous to viscous electron current in graphene. *Nat Commun* **10**, 937 (2019).
- 6 Eyring, H. Viscosity, Plasticity, and Diffusion as Examples of Absolute Reaction Rates. *J. Chem. Phys.* **4**, 283 (1936).
- 7 Konda, S. S. M., Brantley, J. N., Bielawski, C. W. & Makarov, D. E. Chemical reactions modulated by mechanical stress: Extended Bell theory. *J. Chem. Phys.* **135**, 164103 (2011).
- 8 Spikes, H. & Tysoe, W. On the Commonality Between Theoretical Models for Fluid and Solid Friction, Wear and Tribochemistry. *Tribol Lett* **59**, 21 (2015).
- 9 He, F., Yang, X., Bian, Z., Xie, G., Guo, D. & Luo, J. In-Plane Potential Gradient Induces Low Frictional Energy Dissipation during the Stick-Slip Sliding on the Surfaces of 2D Materials. *Small* **15**, 1904613 (2019).

- 10 Shi, B., Gan, X., Yu, K., Lang, H., Cao, X. a., Zou, K. & Peng, Y. Electronic friction and tuning on atomically thin MoS₂. *npj 2D Mater Appl* **6**, 1 (2022).
- 11 Prandtl, L. Ein Gedankenmodell zur kinetischen Theorie der festen Körper. *ZAMM- Journal of Applied Mathematics and Mechanics/Zeitschrift für Angewandte Mathematik und Mechanik* **8**, 85 (1928).
- 12 Tomlinson, G. A. CVI. A molecular theory of friction. *The London, Edinburgh, and Dublin philosophical magazine and journal of science* **7**, 905 (1929).
- 13 Medyanik, S. N., Liu, W. K., Sung, I.-H. & Carpick, R. W. Predictions and Observations of Multiple Slip Modes in Atomic-Scale Friction. *Physical Review Letters* **97**, 136106 (2006).
- 14 Nakamura, J., Wakunami, S. & Natori, A. Double-slip mechanism in atomic-scale friction: Tomlinson model at finite temperatures. *Physical Review B* **72**, 235415 (2005).
- 15 Wang, W., Dietzel, D. & Schirmeisen, A. Single-asperity sliding friction across the superconducting phase transition. *Sci Adv* **6**, eaay0165 (2020).
- 16 Calizo, I., Balandin, A., Bao, W., Miao, F. & Lau, C. Temperature dependence of the Raman spectra of graphene and graphene multilayers. *Nano Lett.* **7**, 2645 (2007).
- 17 Yang, Y., Brenner, K. & Murali, R. The influence of atmosphere on electrical transport in graphene. *Carbon* **50**, 1727 (2012).
- 18 Sang, Y., Dubé, M. & Grant, M. Thermal effects on atomic friction. *Physical Review Letters* **87**, 174301 (2001).
- 19 Yin, Y., Cheng, Z., Wang, L., Jin, K. & Wang, W. Graphene, a material for high temperature devices—intrinsic carrier density, carrier drift velocity and lattice energy. *Scientific reports* **4**, 1 (2014).
- 20 Massicotte, M., Soavi, G., Principi, A. & Tielrooij, K.-J. Hot carriers in graphene—fundamentals and applications. *Nanoscale* **13**, 8376 (2021).

REVIEWERS' COMMENTS

Reviewer #1 (Remarks to the Author):

In the revised manuscript, the authors adopted an extended PT model to explain their results, i.e., they proposed the total potential energy $U_{VG} = U_{ph} + U_{el.VG}$, accounting for phononic and electronic contribution. A similar electronic property fluctuation (EPF) model has been proposed in a previous study [Nano Letters 22, 5, 1889-1896 (2022)], in which the potential energy of the current-carrying friction is considered to be the contribution of electron redistribution and electron transfer energy ΔE_{elec} superimposed on the conventional potential energy, which provides a theoretical framework for prediction of current-carrying or electric field-controlled friction. Further discussion and comparison of the previous current-carrying friction model with the present model will be insightful for readers in this area.

Reviewer #2 (Remarks to the Author):

All my comments are well addressed. No further comments.

Reviewer #3 (Remarks to the Author):

The authors addressed my comments accordingly.

REVIEWERS' COMMENTS

Reviewer #1 (Remarks to the Author):

In the revised manuscript, the authors adopted an extended PT model to explain their results, i.e., they proposed the total potential energy $UVG=U_{ph}+U_{el}.VG$, accounting for phononic and electronic contribution. A similar electronic property fluctuation (EPF) model has been proposed in a previous study [Nano Letters 22, 5, 1889-1896 (2022)], in which the potential energy of the current-carrying friction is considered to be the contribution of electron redistribution and electron transfer energy ΔE_{elec} superimposed on the conventional potential energy, which provides a theoretical framework for prediction of current-carrying or electric field-controlled friction. Further discussion and comparison of the previous current-carrying friction model with the present model will be insightful for readers in this area.

We thank the reviewer for this suggestion. We have carefully read this work and related works. Based on that we have improved our discussion of the contributions of charge transfer between the sliding surfaces. This reference has been added on page 15 of the manuscript with changes tracked, along with a discussion of the results in the context of our measurements.

“Looking at electronic effects of a slightly larger scale, recent work also performed using AFM has associated increased friction with large tribocurrents across the sliding interface of chemically modified semiconductors.³⁸ A study of metal-supported graphene in a conductive AFM demonstrated that current transfer while sliding is associated with higher friction than sliding with no current across the interface, both under an electric field.³⁹ These authors also observed an enhanced friction not fully explained by changes in the electrostatic attraction. In their fully conductive system, they invoke an electronic property fluctuation model that assumes changes in friction are influenced by both an altered electron density at the interface and the promotion of charge transfer between the sliding surfaces due to the electric field. Similar to the present work, these changes are invoked in the context of the PT model, with both the electron density redistribution and charge transfer leading to an enhanced energy barrier for sliding. Our stick slip measurements do suggest a change of the potential energy landscape when charge transfer occurs, but unfortunately the current across the sliding interface cannot be measured in situ due to the electronically floating semiconducting (Si) tip. So, while we cannot definitively say this for our system, another potential explanation is that the measured excess friction is also influenced by the magnitude and rate of charge transferred between the tip and graphene, which change depending on the tip material, graphene’s doping state, and the quantity of induced trapped charges at the interface. The Si tips are more sensitive to these factors than the other two tip types representing the extreme possibilities of charge transfer, and thus charge transfer can be encouraged, discouraged, and rate-controlled depending on the charge density in graphene and the material properties of the countersurface, an AFM tip in our case. The interplay between triboelectric currents and the field effect, particularly in semiconducting contacts, thus presents a route for further experimental investigation into electronic contributions to nanoscale friction.”

Reviewer #2 (Remarks to the Author):

All my comments are well addressed. No further comments.

Thank you so much!

Reviewer #3 (Remarks to the Author):

The authors addressed my comments accordingly.
Thank you so much!